# Strain level microbial detection and quantification with applications to single cell metagenomics

Kaiyuan Zhu[1,2,3], Alejandro A. Schäffer [1], Welles Robinson[1,4], Junyan Xu[1], Eytan Ruppin[1], A. Funda Ergun[3], Yuzhen Ye[3] & S. Cenk Sahinalp [1,3] ✉

Computational identification and quantification of distinct microbes from high throughput sequencing data is crucial for our understanding of human health. Existing methods either use accurate but computationally expensive alignment-based approaches or less accurate but computationally fast alignment-free approaches, which often fail to correctly assign reads to genomes. Here we introduce CAMMiQ, a combinatorial optimization framework to identify and quantify distinct genomes (specified by a database) in a metagenomic dataset. As a key methodological innovation, CAMMiQ uses substrings of variable length and those that appear in two genomes in the database, as opposed to the commonly used fixed-length, unique substrings. These substrings allow to accurately decouple mixtures of highly similar genomes resulting in higher accuracy than the leading alternatives, without requiring additional computational resources, as demonstrated on commonly used benchmarking datasets. Importantly, we show that CAMMiQ can distinguish closely related bacterial strains in simulated metagenomic and real single-cell metatranscriptomic data.

Recent appreciation for the importance of microbes in human health and disease has prompted the generation of many metagenomic HTS (high throughput sequencing) datasets[1]. The increase in available HTS data from human tissues also represents an enormous resource because many of these datasets include reads from tissue-resident microbes, which have been shown to play important roles human disease, including tumorigenesis and the tumor response to therapy[2–8].

The increase in available metagenomic HTS datasets prompted the development of many taxonomic classification and abundance estimation methods. A recent benchmarking study[9] involving a dataset established by Critical Assessment of Metagenome Interpretation (CAMI) challenge and International Microbiome and Multiomics Standards Alliance (IMMSA) provides a comprehensive review of these methods. The study covers 20 taxonomic classifiers including both

alignment-based approaches (such as GATK PathSeq, blastn and MetaPhlAn2[10–12]) as well as alignment-free approaches (such as Kraken, CLARK, KrakenUniq, Centrifuge, and Bracken[13–17]). Below, we provide an overview of the general approaches employed for metagenomic classification methods.

Early approaches for analyzing metagenomic sequencing data were alignment-based and used a reference database. Reads were primarily searched in GenBank[18] through blastn[11] or custom built aligners such as GATK PathSeq[10]. Unfortunately, the growth of HTS data and reference databases has made read search and alignment using blastn or GATK PathSeq computationally infeasible on the largest datasets. For example, a recent study showing that microbial reads from tumors sequenced by The Cancer Genome Atlas (TCGA) can be used to build a classifier for cancer type[19] use the alignment-free approach Kraken[13] due to the large number of samples analyzed. Even

[1]Cancer Data Science Laboratory, National Cancer Institute, National Institutes of Health, Bethesda, MD, USA. [2]Department of Computer Science & Engineering, UC San Diego, La Jolla, CA, USA. [3]Department of Computer Science, Indiana University, Bloomington, IN, USA. [4]Surgery Branch, Center for Cancer Research, National Cancer Institute, National Institutes of Health, Bethesda, MD, USA. ✉e-mail: cenk.sahinalp@nih.gov

though Kraken and other alignment-free tools are faster than the alignment-based tools[20], these alignemnt-free tools are not as accurate. For example, another recent paper on microbial reads from single cell RNA-seq (scRNA-seq) datasets to distinguish cell type specific intracellular microbes from extracellular and contaminating microbes[21] had to use GATK PathSeq because the relatively small number of microbial reads per cell were inadequate for available alignment-free methods to give accurate results. The distinct approaches taken by these two studies exemplify the tradeoffs inherent in the above methodologies.

Alignment-based methods can be sped up substantially by aligning reads to a compressed reference database or to a reference collection of sequences from *marker genes*, which are usually clade-specific, single-copy genes[22,23]. Since marker-gene based methods identify and use only a handful of marker genes on each genome, much of the data goes unused, making taxonomic quantification less accurate. Species with low abundance within the sample may be difficult to identify through marker gene methods because the data may contain few reads originating from the marker genes.

Alignment-free methods typically rely on exact string matching[16,24], or $k$-mer (substrings of length $k$) "matches" to obtain a taxonomic assignment for every read. These methods either assign a read to the lowest taxonomic rank possible (determined by the specificity of the read's substrings, or $k$-mers)[13,25–27], or to a pre-determined taxonomic level, i.e., genus, species, or strain[14,28]. Unlike marker-gene based methods, $k$-mer based applications can use all the input reads[29]. The large memory footprint to maintain the entire $k$-mer profile of each genome, for large values of $k$, can be reduced through hashing or subsampling the $k$-mers[30–33]. In addition to methods based on exact $k$-mer matches, it is also possible to assign metagenomic reads to bacterial genomes by employing sequence-specific features (e.g., short $k$-mer distribution or GC content)[34–38], although methods that employ this approach are typically not very accurate at species level or strain level assignment. These methods, as a result, are typically insufficient for strain-level applications[39], e.g., to identify mixed infections caused by multiple strains of a bacterial species[40–42], to distinguish pathogenic strains from non-pathogenic strains[43], or to track food-borne pathogens[44].

Most of the methods described above and covered in the aforementioned benchmarking study[9] analyze each read without consideration of how the reads are sampled. Provided that the sequence data to be analyzed are genomic DNA, the distribution of HTS reads from a given species or strain should be roughly uniform. This principle is used in several methods for isoform abundance estimation[45–47] and are effective even though the distribution of reads across an isoform may not be uniform in practice. In the context of metagenomic abundance estimation, however, the uniform coverage principle is under-utilized. One exception is the network flow based approach, utilized, for example, by ref. 48, which does take into account the uniform coverage—however, it is relatively slow due to the hardness of the underlying algorithmic problem. Another method that utilizes the near uniformity across $k$-mers within a genome is ref. 15, which runs faster but also is less accurate.

In addition to the metagenomic species identification and quantification methods summarized above, there are also tools to determine the likely presence of a long genomic sequence (e.g., the complete or partial genome of a bacterial species) in a given metagenomic sample[49–53]. Even though these tools solve an entirely different problem, methodologically they are similar to the $k$-mer based metagenomic identification and quantification tools such as refs. 13,14, in the sense that they build a succinct index on the database, which is comprised of the metagenomic read collection, and they query this index without explicit alignment. However, because of their design parameters, these tools can not perform abundance estimation.

In this paper, we describe CAMMiQ (Combinatorial Algorithms for Metagenomic Microbial Quantification), a computational approach to maintain/manage a collection of $m$ (bacterial) genomes $\mathcal{S} = \{s_1, \ldots, s_m\}$, each assembled into one or more strings/contigs, representing a species, a particular strain of a species, or any other taxonomic rank. CAMMiQ constructs a data structure, which can answer queries of the following form: given a set $\mathcal{Q}$ of HTS reads obtained from a mixture of genomes or transcriptomes, each from $\mathcal{S}$, identify the genomes in $\mathcal{Q}$, and, in case the reads are genomic, compute their relative abundances. Our data structure is very efficient in terms of its empirical querying time and is shown to be very accurate on simulations for which the ground truth answers are known. The distinctive feature of our data structure is its utilization of substrings that are present in at most $c$ genomes ($c > 1$) in $\mathcal{S}$; in this paper, we focus on $c = 2$, which we call doubly-unique substrings. CAMMiQ is thus different from available methods which set $c = 1$ to compare genomes via their shortest unique substrings[54,55], or perform metagenomic analysis by employing $k$-mers unique to each genome[13–15]. By considering substrings that are present in $c = 2$ (or possibly more) genomes, CAMMiQ utilizes a higher proportion of reads and can accurately identify genomes at subspecies/strain level. The choice of $c = 2$ is sufficiently powerful for the datasets we considered. However, our approach can be generalized for any fixed value of $c \geq 2$. Another distinctive feature of our data structure is its use of the variable length substrings—rather than fixed length $k$-mers. Because any extension of a shortest unique substring is also unique, CAMMiQ only maintains the shortest of these overlapping unique substrings to maximize utility. By being flexible about substring length, CAMMiQ potentially has a a larger selection of substrings from which to choose; because it utilizes the shortest unique substrings, it maximizes possible coverage. To assign each read in $\mathcal{Q}$ that includes an almost-unique substring (i.e., a string present in at most $c$ genomes) to a genome, our data structure solves an integer linear program (ILP) - that simultaneously infers which genomes are present in $\mathcal{Q}$ and, if the reads are genomic, the relative abundances of the identified genomes. Specifically, the objective of the ILP is to identify a set of genomes in which the coverage of the almost-unique substrings in each genome is (approximately) uniform. Our final contribution is a set of conditions sufficient to identify and quantify genomes in a query correctly, through the use of unique substrings/ $k$-mers, provided the reads are error-free. Although this is a purely theoretical result, to the best of our knowledge it has not been applied to metagenomic data analysis, and is valid for CAMMiQ for the case $c = 1$ and other unique substring based methods such as CLARK and KrakenUniq. Setting $c = 2$ for CAMMiQ is advised for cases where these conditions are not met. On the experimental side, we show that CAMMiQ is not only much faster but also more accurate than the mapping based GATK PathSeq, which, as mentioned earlier, was used on scRNA-seq data obtained from monocyte-derived dendritic cells (moDCs) infected with distinct *Salmonella* strains[21]—where accuracy was the top priority. The application to single-cell data is important because in studies of the human microbiome, it is of interest to know which cells are infected with which microbial strains, especially to distinguish between benign commensals and pathogenic variants of bacteria such as E. coli. Using current sequencing technologies, single-cell nucleotide data are primarily RNAseq rather than DNAseq, which is why we focus on an RNAseq case study. Returning to the established problem of analyzing bulk DNAseq data, we demonstrate the comparative advantage of CAM–MiQ against the top performing alignment based and alignment free metagenomic classification methods according to the above-mentioned benchmarking study[9] on the very same (CAMI and IMMSA) dataset. We additionally show that CAMMiQ is uniquely capable of handling particularly challenging microbial strains we derived from the NCBI RefSeq database.

## Results

Below, we first give a brief overview of CAMMiQ algorithm. Then we describe the index data sets, simulated and real query sets, as well as the alternative computational methods we used to benchmark CAMMiQ's performance. We next demonstrate CAMMiQ's comparative accuracy performance against alternative metagenomic analysis methods on the two species level data sets we have: the first is the CAMI and IMMSA benchmark (i.e., species-level-all) index dataset and the second is the species-level-bacteria index dataset. For these two datasets, we not only provide accuracy figures for the tools benchmarked but also the computational resources they use. Additionally, we demonstrate the maximum potential advantage that could be offered by CAMMiQ through its use of doubly-unique, variable length substrings on our species-level-bacteria index dataset. We then demonstrate CAMMiQ's performance on our strain-level index dataset. Finally, we demonstrate CAMMiQ's performance on real metatranscriptomic query sets through its use of our subspecies-level index dataset. The results of CAMMiQ in this setup was compared against that of the GATK PathSeq tool[10] which was utilized by the original study on this data set[21], as well as blastn method[11], which possibly offers the most accurate (albeit slow) approach for the relevant purpose.

### Overview of CAMMiQ indexing and querying procedure

As per a typical metagenomic classification or profiling tool, CAMMiQ involves two steps, namely, index construction and query. In the index construction step, CAMMiQ is given a set $\mathcal{S} = \{s_i\}_{i=1}^m$ of $m$ genomes or contigs, each labeled with an ID representing the taxonomy of that genome. We call $\mathcal{S}$ an *index dataset* below. By the end of this step, CAMMiQ returns the collection of sparsified shortest unique substrings and shortest doubly-unique substrings on each genome $s_i$ in $\mathcal{S}$ in a compressed binary format, and other meta information involving the input index dataset, which jointly composing its index on the dataset. CAMMiQ reuses its index in the next query step.

In the query step, CAMMiQ is given a collection of reads $\mathcal{Q} = \{r_j\}_{j=1}^n$ of varying length, and identifies a set of genomes $\mathcal{A} = \{s_1, \cdots, s_a\} \subset \mathcal{S}$ and their respective abundances $p_1, \cdots, p_a$ that "best explain" $\mathcal{Q}$ efficiently. We call $\mathcal{Q}$ a *query* or *query set* below. Depending on specific applications, a user can select to return (i) $\mathcal{A}_1 \subseteq \mathcal{S}$, the set of genomes such that each includes at least one shortest unique substring that also occur in some read $r_j$ in the query $\mathcal{Q}$; (ii) $\mathcal{A}_2 \subseteq \mathcal{S}$, the smallest subset of genomes in $\mathcal{S}$ which include all shortest unique and doubly-unique substrings that also occur in some read $r_j \in \mathcal{Q}$; or $\mathcal{A}_3 \subseteq \mathcal{S}$, the smallest subset of $\mathcal{S}$ which again include all shortest unique and doubly-unique substrings that also occur in some read $r_j \in \mathcal{Q}$, with the additional constraint that the "coverage" of these substrings in each genome $s_i \in \mathcal{A}_3$ is roughly uniform. In the last case CAMMiQ also computes the relative abundance of each genome $s_i$ in $\mathcal{A}_3$.

For all three query types, CAMMiQ first identifies for each read $r_j$ all unique and doubly-unique substrings it includes; it then assigns $r_j$ to the one or two genomes from which these substrings possibly originate. To compute $\mathcal{A}_1$, CAMMiQ simply returns the collection of genomes receiving at least one read assignment. To compute $\mathcal{A}_2$, CAMMiQ solves a hitting set problem though an ILP, where genomes form the universe of items, and indexed strings that appear in query reads form the sets of items to be hit. To compute $\mathcal{A}_3$, CAMMiQ solves the combinatorial optimization problem that asks to minimize the variance among the number of reads assigned to each indexed substring of each genome, again through an ILP. The solution indicates the set of genomes in $\mathcal{A}_3$ along with their respective abundances.

### Datasets

To evaluate the overall performance of CAMMiQ, we have performed four sets of experiments, each with a distinct index dataset (all based on NCBI's RefSeq database[56]) and a distinct collection of queries.

(i) The first, species-level-all dataset is the most comprehensive index dataset, which includes one complete genome from each bacterial, viral and archaeal species from NCBI's RefSeq database, resulting in a total of $m = 16,418$ genomes. This dataset is established for the CAMI and IMMSA repository used in recent benchmarking studies of metagenomics classification and profiling tools[9,57]. There are 16 query sets from this repository used in these two studies, 8 from CAMI and 8 from IMMSA. Notably, both CAMI and IMMSA query sets include genomes that are not present in the species-level-all index dataset. In fact, the CAMI query sets include only a small porportion of genomes from the index dataset - the majority of the reads in these queries represent unknown species or simulated strains "evolved" from known species that are not in the species-level-all index dataset. See Supplementary Notes 5.4.1 and 5.4.2 for a detailed description of these queries. We used these query sets to demonstrate the comparative performance of CAMMiQ against the best performing methods according to ref. 9, namely Kraken2[58], KrakenUniq[15], CLARK[14], Centrifuge[16], and Bracken[17]; please see Supplementary Note 6 for the specific parameters and setup used for each of these tools. Since genomes in the query sets may not be all included in the index dataset, we employed query type $\mathcal{A}_2$ to evaluate CAMMiQ's performance against the aforementioned tools.

(ii) We compiled our next, species-level-bacteria index dataset to evaluate the species level performance of CAMMiQ, this time across one representative complete genome from each of the $m = 4122$ bacterial species from (an earlier version of) NCBI's RefSeq. This index dataset enabled us to measure the performance of CAMMiQ's type $\mathcal{A}_3$ queries against the tools mentioned above plus MetaPhlAn2[12], a marker-gene based profiling tool. We simulated 14 query sets for this experiment with varying levels of "difficulty" across the genomes. These include 10 challenging (marked Least) and 4 easier queries (marked Random). See Supplementary Note 5.4.3 and Supplementary Fig. 2 for a detailed description of these queries.

(iii) Our next strain-level index dataset is smaller: it includes the complete set of $m = 614$ human gut related bacterial strains from ref. 59 for the purpose of evaluating CAMMiQ's strain level performance. We again employed type $\mathcal{A}_3$ queries of CAMMiQ to compare it against the above-mentioned tools. We simulated 4 queries for this index dataset with varying levels of "difficulty". See Supplementary Note 5.5 for details.

(iv) We finally evaluated CAMMiQ on a dataset from another study[60] which involved metatranscriptomic reads from 262 single human immune cells (monocyte-derived dendritic cells, moDCs) deliberately infected with two distinct strains of the intracellular bacterium *Salmonella* enterica and 80 uninfected cells used as negative controls. A recent study[21] applied the GATK PathSeq tool[10] to these metatranscriptomic read sets to validate the presence of *Salmonella* genus in each cell. To demonstrate CAMMiQ's ability to distinguish cells infected with specific strains of *Salmonella* in time much faster than GATK PathSeq, we applied its query types $\mathcal{A}_1$ and $\mathcal{A}_2$ to these metatranscriptomic read sets. Since these are not genomic reads, our query type $\mathcal{A}_3$ could not be used. The index dataset we used for these queries are at the subspecies-level; it consists of $m = 3395$ complete bacterial genomes, where each species is represented by a handful of strains. This index dataset was generated to reduce the sampling bias observed in the RefSeq database, which, e.g., includes more than 300 strains from the genus *Salmonella*. CAMMiQ's accuracy was compared mainly against PathSeq (a mapping based, thus relatively slow method) for this experiment since PathSeq was the preferred method of the original study due to its high levels of accuracy. Further details on the real query sets can be found in Supplementary Note 5.6.

**Table 1 | Synthetic and real bacterial read sets used to benchmark** `CAMMiQ`**'s performance against the best performing metagenomic classification and abundance estimation tools**

| Query set | Index dataset & Query type | Read length (L) | Num. reads (n) | Err Rate |
|---|---|---|---|---|
| IMMSA-buccal-12 | `species-level-all` | 100 | 0.6M | 0.001 |
| IMMSA-citypark-48 | $\mathcal{A}_2$ | 100 | 1.2M | 0.001 |
| IMMSA-gut-20 | | 100 | 0.5M | 0.001 |
| IMMSA-house-30 | | 100 | 0.75M | 0.001 |
| IMMSA-house-20 | | 100 | 0.5M | 0.001 |
| IMMSA-soil-50 | $\mathcal{A}_2$ | 100 | 2.5M | 0.001 |
| IMMSA-simBA-525 | | 100 | 5.7M | 0.001 |
| IMMSA-nycsm-20 | | 100 | 0.5M | 0.001 |
| CAMI-LC-1 | | 150 | 99.8M | |
| CAMI-MC-1 | | 150 | 99.8M | |
| CAMI-MC-2 | | 150 | 99.8M | |
| CAMI-HC-1 | | 150 | 99.8M | Unknown |
| CAMI-HC-2 | | 150 | 99.8M | |
| CAMI-HC-3 | | 150 | 99.8M | |
| CAMI-HC-4 | | 150 | 99.8M | |
| CAMI-HC-5 | | 150 | 99.8M | |
| Least-20-uniform-1 | `species-level-bacteria` | 100 | 4.8M | 0 |
| Least-20-uniform-2 | $\mathcal{A}_3$ | 100 | 4.8M | 0.01 |
| Least-20-uniform-3 (uneven) | | 100 | 4.8M | 0.01 |
| Least-quantifiable-20-uniform-1 | | 100 | 5.0M | 0 |
| Least-quantifiable-20-uniform-2 | | 100 | 5.0M | 0.01 |
| Least-quantifiable-20-uniform-3 (uneven) | | 100 | 5.0M | 0.01 |
| Least-20-genera-uniform-1 | | 100 | 4.0M | 0 |
| Least-20-genera-uniform-2 | $\mathcal{A}_3$ | 100 | 4.0M | 0.01 |
| Least-20-genera-uniform-3 (uneven) | | 100 | 4.0M | 0.01 |
| Least-20-genera-lognormal | | 100 | 4.0M | 0.01 |
| Random-20-uniform | | 100 | 4.4M | 0.01 |
| Random-20-lognormal | | 100 | 5.0M | 0.01 |
| Random-20-lognormal-a.g. | | 100 | 1.1M | 0.01 |
| Random-100-uniform | | 100 | 21.5M | 0.006 |
| HumanGut-least-25 | `strain-level` | 100 | 2.0M | 0.01 |
| HumanGut-random-100-1 | | 100 | 8.0M | 0.01 |
| HumanGut-random-100-2 | | 125 | 8.0M | 0.01 |
| HumanGut-all | | 100 | 20.0M | 0.01 |
| Filtered-scRNA-seq | `subspecies-level` | 66.4(13.6) | 8.5M | Unknown |
| | $\mathcal{A}_1, \mathcal{A}_2$ & $\mathcal{A}_3$ | | | |

The first dataset is comprised of the entire collection of 8 IMMSA queries and 8 CAMI queries from an earlier benchmark. Since these queries were not sampled from an available index dataset, we compiled a comprehensive `species-level-all` index dataset consisting of 16,418 distinct bacterial, viral and archaeal species from RefSeq. We sampled our second query set from the genomes of another `species-level-bacteria` index dataset we complied which consists of 4122 distinct bacterial species from RefSeq. We similarly sampled our third collection of queries from the genomes of our `strain-level` index dataset consisting of 614 (possibly incompletely assembled) human gut bacteria from an earlier study. The final collection of queries consisted of real single cell RNA-seq reads (with average length 66.4bp and standard deviation 13.6bp) sequenced from 342 immune cells infected with *Salmonella enterica*; for this, we compiled a `subspecies-level` index dataset comprised of a selection of 3395 bacterial genomes from 2753 distinct species. Details on our query sets and the corresponding index datasets can be found in Supplementary Note 5.

A summary of data sets used in our experiments can be found in Table 1. Additional details on the four index datasets can be found in Supplementary Notes 5.1–5.3. As will be demonstrated, `CAMMiQ`'s performance on these query sets is superior to all alternatives in almost all scenarios we tested.

**Precision and recall in read classification across all species level queries**

We tested `CAMMiQ`'s species level performance on both CAMI and IMMSA (i.e., `species-level-all`) and `species-level-bacteria` data sets, and compared it against the best performing alternatives according to ref. [9]. Results based on CAMI and IMMSA are summarized in Table 2; results based on `species-level-bacteria` data set are summarized in Table 3.

Perhaps the most widely-used performance measures to benchmark metagenomic classifiers are the proportion of reads correctly assigned to a genome among (i) the set of reads assigned to some genome, i.e., precision, and (ii) the full set of reads in the query, i.e., recall[14]. In Table 2, panel A, as well as Table 3, panel A, we report the selected tools precision in read classification. Then, in Table 2 panel B and Table 3, panel B, we report these tools' recall in read classification.

Note that the above tables do not report the read classification precision and recall values for MetaPhlAn2. This is partially due to MetaPhlAn2's use of an index based on a very different (pre-determined) and much smaller database of marker genes. As a consequence, MetaPhlAn2 assigns very few reads to the marker genes in its database and thus appears to have very low recall (and possibly higher precision). This would not accurately reflect

**Table 2 | Performance evaluation of** `CAMMiQ`**, Kraken2, KrakenUniq, CLARK, Centrifuge, and Bracken on CAMI and IMMSA benchmark queries against the** `species-level-all` **index dataset**

| Performance measure | Query set | CAMMiQ | Kraken2 | KrakenUniq | CLARK | Centrifuge | Bracken |
|---|---|---|---|---|---|---|---|
| A. Classification Precision | IMMSA-buccal-12 | 0.755 | 0.431 | 0.649 | 0.649 | 0.063 | N/A |
| | IMMSA-citypark-48 | 0.895 | 0.802 | 0.890 | 0.888 | 0.671 | |
| | IMMSA-gut-20 | 0.793 | 0.472 | 0.759 | 0.759 | 0.168 | |
| | IMMSA-house-30 | 0.841 | 0.624 | 0.784 | 0.783 | 0.271 | |
| | IMMSA-house-20 | 0.799 | 0.594 | 0.756 | 0.756 | 0.391 | |
| | IMMSA-soil-50 | 0.902 | 0.775 | 0.884 | 0.884 | 0.693 | |
| | IMMSA-simBA-525 | 0.954 | 0.737 | 0.903 | 0.905 | 0.553 | |
| | IMMSA-nycsm-20 | 0.749 | 0.582 | 0.729 | 0.727 | 0.362 | |
| B. Classification Recall | IMMSA-buccal-12 | 0.559 | 0.215 | 0.551 | 0.546 | 0.048 | N/A |
| | IMMSA-citypark-48 | 0.751 | 0.371 | 0.779 | 0.778 | 0.572 | |
| | IMMSA-gut-20 | 0.529 | 0.189 | 0.550 | 0.547 | 0.114 | |
| | IMMSA-house-30 | 0.697 | 0.311 | 0.680 | 0.678 | 0.207 | |
| | IMMSA-house-20 | 0.636 | 0.277 | 0.641 | 0.639 | 0.309 | |
| | IMMSA-soil-50 | 0.748 | 0.363 | 0.763 | 0.762 | 0.572 | |
| | IMMSA-simBA-525 | 0.798 | 0.417 | 0.786 | 0.784 | 0.452 | |
| | IMMSA-nycsm-20 | 0.582 | 0.223 | 0.568 | 0.566 | 0.265 | |
| C. Identification Recall | IMMSA-buccal-12 | 1.0 | 0.818 | 0.818 | 0.818 | 1.0 | 1.0 |
| | IMMSA-citypark-48 | 0.979 | 0.915 | 0.936 | 0.936 | 0.979 | 0.979 |
| | IMMSA-gut-20 | 0.895 | 0.789 | 0.789 | 0.789 | 0.947 | 0.895 |
| | IMMSA-house-30 | 0.963 | 0.889 | 0.889 | 0.926 | 0.963 | 1.0 |
| | IMMSA-house-20 | 0.895 | 0.789 | 0.789 | 0.789 | 0.895 | 0.895 |
| | IMMSA-soil-50 | 0.958 | 0.917 | 0.938 | 0.938 | 0.979 | 0.958 |
| | IMMSA-simBA-525 | 0.933 | 0.913 | 0.923 | 0.933 | 0.935 | 0.939 |
| | IMMSA-nycsm-20 | 0.895 | 0.789 | 0.789 | 0.789 | 0.895 | 0.947 |
| | CAMI-LC-1 | 1.0 | 1.0 | 1.0 | 1.0 | 1.0 | 1.0 |
| | CAMI-MC-1 | 0.864 | 0.864 | 0.818 | 0.818 | 0.864 | 0.818 |
| | CAMI-MC-2 | 0.955 | 0.955 | 0.909 | 0.909 | 0.909 | 0.955 |
| | CAMI-HC-1 | 0.900 | 0.850 | 0.900 | 0.950 | 0.900 | 0.950 |
| | CAMI-HC-2 | 0.900 | 0.850 | 0.900 | 0.900 | 0.950 | 0.900 |
| | CAMI-HC-3 | 0.950 | 0.850 | 0.950 | 0.950 | 0.950 | 0.950 |
| | CAMI-HC-4 | 0.950 | 0.850 | 0.950 | 0.950 | 0.950 | 0.900 |
| | CAMI-HC-5 | 0.950 | 0.850 | 0.950 | 0.950 | 0.950 | 0.950 |
| D. Identification Precision | IMMSA-buccal-12 | 0.162 | 0.006 | 0.098 | 0.086 | 0.110 | 0.007 |
| | IMMSA-citypark-48 | 0.426 | 0.178 | 0.303 | 0.268 | 0.333 | 0.122 |
| | IMMSA-gut-20 | 0.145 | 0.009 | 0.111 | 0.098 | 0.122 | 0.014 |
| | IMMSA-house-30 | 0.310 | 0.027 | 0.198 | 0.166 | 0.234 | 0.033 |
| | IMMSA-house-20 | 0.254 | 0.022 | 0.172 | 0.147 | 0.189 | 0.023 |
| | IMMSA-soil-50 | 0.331 | 0.175 | 0.281 | 0.263 | 0.301 | 0.119 |
| | IMMSA-simBA-525 | 0.842 | 0.532 | 0.830 | 0.820 | 0.827 | 0.564 |
| | IMMSA-nycsm-20 | 0.168 | 0.013 | 0.109 | 0.088 | 0.113 | 0.024 |
| | CAMI-LC-1 | 0.010 | 0.001 | 0.004 | 0.004 | 0.003 | 0.001 |
| | CAMI-MC-1 | 0.026 | 0.006 | 0.019 | 0.018 | 0.016 | 0.005 |
| | CAMI-MC-2 | 0.036 | 0.006 | 0.027 | 0.024 | 0.021 | 0.006 |
| | CAMI-HC-1 | 0.014 | 0.004 | 0.012 | 0.012 | 0.010 | 0.005 |
| | CAMI-HC-2 | 0.014 | 0.004 | 0.011 | 0.011 | 0.011 | 0.005 |
| | CAMI-HC-3 | 0.014 | 0.004 | 0.012 | 0.011 | 0.011 | 0.005 |
| | CAMI-HC-4 | 0.014 | 0.004 | 0.012 | 0.011 | 0.011 | 0.005 |
| | CAMI-HC-5 | 0.014 | 0.004 | 0.012 | 0.012 | 0.011 | 0.005 |

Classification Precision: the proportion of reads correctly assigned to a genome among the set of reads assigned to some genome (correctly or incorrectly). Classification Recall: the proportion of reads correctly assigned to a genome among the total number of reads in a query. Identification Recall: the number of correctly identified genomes (true positives) over the total number of genomes existing in each query. Identification Precision: the number of correctly identified genomes (true positives) over the number of genomes with abundance > 0.0001 reported by each software tool in each query.

**Table 3 | Performance evaluation of** `CAMMiQ`**, Kraken2, KrakenUniq, CLARK, Centrifuge, Bracken and MetaPhlAn2 on the 14** `species-level-bacteria` **queries**

| Performance measure | Query set | CAMMiQ | Kraken2 | KrakenUniq | CLARK | Centrifuge | Bracken | MetaPhlAn2 |
|---|---|---|---|---|---|---|---|---|
| A. Classification | Least-20-uniform-1 | 1.0 | 0.014 | 0.952 | 1.0 | 1.0 | N/A | N/A |
| Precision | Least-20-uniform-2 | 0.974 | 0.001 | 0.009 | 0.006 | 0.997 | | |
| | Least-quantifiable-20-uniform-1 | 1.0 | 0.043 | 0.936 | 1.0 | 1.0 | | |
| | Least-quantifiable-20-uniform-2 | 0.989 | 0.012 | 0.086 | 0.064 | 0.999 | | |
| | Least-20-genera-uniform-1 | 1.0 | 0.027 | 0.978 | 1.0 | 1.0 | | |
| | Least-20-genera-uniform-2 | 0.993 | 0.005 | 0.047 | 0.028 | 0.999 | | |
| | Least-20-genera-lognormal | 0.993 | 0.005 | 0.049 | 0.033 | 0.999 | | |
| | Random-20-uniform | 0.997 | 0.783 | 0.981 | 0.994 | 0.999 | | |
| | Random-20-lognormal | 0.998 | 0.933 | 0.991 | 0.995 | 0.999 | | |
| | Random-20-lognormal-a.g.* | 0.998 | 0.900 | 0.998 | 0.997 | 1.0 | | |
| | Random-100-uniform | 0.998 | 0.968 | 0.965 | 0.997 | 0.999 | | |
| B. Classification | Least-20-uniform-1 | 0.358 | < 0.001 | < 0.001 | < 0.001 | < 0.001 | N/A | N/A |
| Recall | Least-20-uniform-2 | 0.317 | < 0.001 | < 0.001 | < 0.001 | 0.004 | | |
| | Least-quantifiable-20-uniform-1 | 0.635 | < 0.001 | 0.001 | 0.001 | 0.001 | | |
| | Least-quantifiable-20-uniform-2 | 0.557 | < 0.001 | 0.001 | 0.001 | 0.013 | | |
| | Least-20-genera-uniform-1 | 0.722 | < 0.001 | < 0.001 | < 0.001 | < 0.001 | | |
| | Least-20-genera-uniform-2 | 0.640 | < 0.001 | < 0.001 | < 0.001 | 0.008 | | |
| | Least-20-genera-lognormal | 0.627 | < 0.001 | < 0.001 | < 0.001 | 0.010 | | |
| | Random-20-uniform | 0.862 | 0.586 | 0.855 | 0.851 | 0.864 | | |
| | Random-20-lognormal | 0.893 | 0.716 | 0.878 | 0.874 | 0.890 | | |
| | Random-20-lognormal-a.g. | 0.830 | 0.759 | 0.883 | 0.880 | 0.888 | | |
| | Random-100-uniform | 0.903 | 0.763 | 0.867 | 0.865 | 0.878 | | |
| C. Num. Correctly | Least-20-uniform-1 | 20/20 | 15/1519 | 17/17 | 18/18 | 18/18 | 11/24 | ≤13/7 |
| Identified Genomes | Least-20-uniform-2 | 20/28 | 7/2340 | 15/157 | 15/1157 | 17/110 | 11/62 | ≤13/8 |
| | Least-20-uniform-3 | 20/29 | 7/2278 | 13/156 | 14/1127 | 16/109 | 12/59 | ≤13/7 |
| | Least-quantifiable-20-uniform-1 | 20/20 | 18/2090 | 19/19 | 20/20 | 19/19 | 18/58 | ≤18/17 |
| | Least-quantifiable-20-uniform-2 | 20/27 | 17/2349 | 19/351 | 20/1518 | 19/114 | 17/149 | ≤18/17 |
| | Least-quantifiable-20-uniform-3 | 20/25 | 16/2205 | 19/370 | 20/1502 | 19/119 | 17/137 | ≤18/16 |
| | Least-20-genera-uniform-1 | 20/20 | 18/1978 | 18/18 | 18/18 | 19/19 | 16/44 | ≤12/19 |
| | Least-20-genera-uniform-2 | 20/24 | 14/2644 | 18/348 | 18/1843 | 18/115 | 16/108 | ≤12/20 |
| | Least-20-genera-uniform-3 | 20/23 | 13/2601 | 18/356 | 18/1855 | 18/122 | 15/108 | ≤12/18 |
| | Least-20-genera-lognormal | 20/33 | 11/2695 | 17/357 | 17/1817 | 17/90 | 13/119 | ≤12/20 |
| | Random-20-uniform | 20/20 | 18/117 | 20/21 | 20/22 | 20/26 | 18/152 | ≤11/20 |
| | Random-20-lognormal | 20/21 | 20/46 | 20/22 | 20/23 | 20/28 | 20/49 | ≤9/17 |
| | Random-20-lognormal-a.g.* | 20/20 | 19/34 | 20/23 | 20/23 | 20/22 | 19/32 | ≤13/14 |
| | Random-100-uniform | 100/100 | 99/101 | 99/99 | 99/99 | 100/103 | 100/107 | ≤76/88 |
| D. L1 Err. | Least-20-uniform-1 | 0.0790 | 1.0000 | 0.9999 | 0.8846 | 0.8043 | 1.0687 | 0.7186 |
| | Least-20-uniform-2 | 0.0929 | 1.0000 | 0.9999 | 0.9940 | 0.9756 | 0.9996 | 0.7200 |
| | Least-20-uniform-3 | 0.1889 | 1.0000 | 0.9999 | 0.9939 | 0.9572 | 0.8635 | 0.7556 |
| | Least-quantifiable-20-uniform-1 | 0.0375 | 0.9994 | 0.9988 | 0.5774 | 0.5782 | 0.1450 | 0.5247 |

**Table 3 (continued) | Performance evaluation of** `CAMMiQ`**, Kraken2, KrakenUniq, CLARK, Centrifuge, Bracken and MetaPhlAn2 on the 14** `species-level-bacteria` **queries**

| Performance measure | Query set | CAMMiQ | Kraken2 | KrakenUniq | CLARK | Centrifuge | Bracken | MetaPhlAn2 |
|---|---|---|---|---|---|---|---|---|
| | Least-quantifiable-20-uniform-2 | 0.0278 | 0.9994 | 0.9989 | 0.9356 | 0.8794 | 0.2082 | 0.5507 |
| | Least-quantifiable-20-uniform-3 | 0.0670 | 0.9995 | 0.9990 | 0.9371 | 0.9671 | 0.1823 | 0.5182 |
| | Least-20-genera-uniform-1 | 0.0626 | 1.0000 | 0.9995 | 0.7156 | 0.1018 | 0.1953 | 0.9153 |
| | Least-20-genera-uniform-2 | 0.0591 | 1.0000 | 0.9996 | 0.9716 | 0.5574 | 0.2240 | 0.9533 |
| | Least-20-genera-uniform-3 | 0.0670 | 1.0000 | 0.9996 | 0.9759 | 0.7119 | 0.2592 | 0.9797 |
| | Least-20-genera-lognormal | 0.0439 | 0.9999 | 0.9995 | 0.9669 | 0.6592 | 0.2757 | 1.0071 |
| | Random-20-uniform | 0.0113 | 0.4139 | 0.1446 | 0.2000 | 0.0026 | 0.2393 | 0.6831 |
| | Random-20-lognormal | 0.0038 | 0.2843 | 0.1217 | 0.1472 | 0.0107 | 0.0674 | 1.7367 |
| | Random-20-lognormal-a.g.* | 0.1262 | 0.2412 | 0.1174 | 0.1831 | 0.1252 | 0.1293 | 0.5578 |
| | Random-100-uniform | 0.0096 | 0.2365 | 0.1328 | 0.2038 | 0.0604 | 0.0336 | 0.7106 |
| E. L2 Err. | Least-20-uniform-1 | 0.0215 | 0.2578 | 0.2578 | 0.2266 | 0.3625 | 0.5230 | 0.2294 |
| | Least-20-uniform-2 | 0.0267 | 0.2578 | 0.2578 | 0.2569 | 0.3837 | 0.4648 | 0.2240 |
| | Least-20-uniform-3 | 0.0716 | 0.2578 | 0.2578 | 0.2568 | 0.3613 | 0.3428 | 0.2499 |
| | Least-quantifiable-20-uniform-1 | 0.0105 | 0.2438 | 0.2437 | 0.1725 | 0.1955 | 0.0461 | 0.1340 |
| | Least-quantifiable-20-uniform-2 | 0.0082 | 0.2438 | 0.2437 | 0.2296 | 0.2058 | 0.0661 | 0.1407 |
| | Least-quantifiable-20-uniform-3 | 0.0200 | 0.2438 | 0.2437 | 0.2300 | 0.2168 | 0.0595 | 0.1385 |
| | Least-20-genera-uniform-1 | 0.0173 | 0.2502 | 0.2501 | 0.1941 | 0.0524 | 0.1020 | 0.2373 |
| | Least-20-genera-uniform-2 | 0.0179 | 0.2502 | 0.2501 | 0.2442 | 0.1598 | 0.1038 | 0.2423 |
| | Least-20-genera-uniform-3 | 0.0267 | 0.2502 | 0.2501 | 0.2450 | 0.1828 | 0.1132 | 0.2440 |
| | Least-20-genera-lognormal | 0.0126 | 0.3111 | 0.3110 | 0.3002 | 0.2334 | 0.0900 | 0.3348 |
| | Random-20-uniform | 0.0034 | 0.1665 | 0.0703 | 0.0677 | 0.0008 | 0.1483 | 0.1922 |
| | Random-20-lognormal | 0.0013 | 0.1205 | 0.0811 | 0.0677 | 0.0056 | 0.0275 | 0.6180 |
| | Random-20-lognormal-a.g. | 0.0675 | 0.1150 | 0.0916 | 0.0963 | 0.0671 | 0.0913 | 0.1895 |
| | Random-100-uniform | 0.0015 | 0.0391 | 0.0302 | 0.0308 | 0.0177 | 0.0052 | 0.0836 |

Classification Precision: the proportion of reads correctly assigned to a genome among the set of reads assigned to some genome (correctly or incorrectly). Classification Recall: the proportion of reads correctly assigned to a genome among the total number of reads in a query. Number of correctly identified genomes: separated by "/", we report respectively the number of correctly identified genomes (true positives) and the total number of genomes returned by each software tool; the one exception is MetaPhlAn2, for which we consider a genome to have been correctly identified even if only its genus (but not the species) is reported, and give the total number of genomes in identified genera as true positives. L1 (or L2) error: the L1 (or L2) distance between the true relative abundance values (between 0 and 1) and the predicted abundance values for each genome in the query (i.e., positives). We made an exception for MetaPhlAn2, where we measured the genus level L1 and L2 distances. Note that we converted the true abundance values reported by Kraken2, KrakenUniq and CLARK by dividing the predicted abundance value for each genome by its length and then normalizing these values by the total abundance value of all genomes.

*10% reads in the query Random-20-lognormal-a.g. are from a genome excluded from the index; any assignment of these reads are necessarily incorrect by all tools except MetaPhlAn2 - whose pre-built index includes this genome.

MetaPhlAn2's performance since unlike the other tools we benchmarked, MetaPhlAn2 does not aim to assign as many reads reads to genomes correctly but rather aims to identify distinct genomes in a metagenomic sample; see Supplementary Note 6 for details. Additionally note that for our bookkeeping purposes, any read assigned to a taxonomic level strictly higher than the species level by Kraken2, KrakenUniq, and Centrifuge is considered to be not assigned. This likely increases their reported precision but may decrease their recall.

In all our species level tests, we used `CAMMiQ`'s default parameter settings of $L_{min} = 26$ and $L_{max} = 50$ to compare it against Kraken2, KrakenUniq, Bracken, CLARK, and Centrifuge, all using $k$-mer length of 26; see Supplementary Note 6 for details on parameter settings. Results based on alternative parameter settings can also be found in

Supplementary Note 8 and in particular Supplementary Table 6. In all of these experiments, we used the same collection of genomes for establishing the index for each of the five tools (with the exception of MetaPhlAn2, which uses its own predetermined index): the results in Table 3 are based on our `species-level-bacteria` index dataset and the results in Table 2 are based on our `species-level-all` index dataset.

Compared with the `species-level-bacteria` queries which are composed of highly similar genomes, the CAMI and IMMSA queries are, in principle, less challenging since reads that did not get mapped to a unique genome were excluded from these queries at the time they were complied[57]. Even though the RefSeq database has been significantly updated since these queries were complied, almost all reads in these queries still map to a unique genome.

Having said that, reads in these queries may originate from genomes outside of the `species-level-all` index dataset - including plasmids from these species that have not been indexed. It is entirely possible that such reads may include one or more unique or doubly-unique substring(s) indexed by `CAMMiQ`, and thus be assigned to the wrong genome.

As can be seen in Table 2, panel A, `CAMMiQ` offered the best precision in read classification for all IMMSA queries; interestingly the precision values for Centrifuge was much lower than the alternatives. `CAMMiQ` was arguably the best on the recall in read classification on the IMMSA queries as well, as can be seen in Table 2, panel B. However, reads that originate from genomes outside of the index database were likely not utilized by `CAMMiQ`, reducing its comparative advantage against, KrakenUniq and CLARK, which may still assign such reads to a genome; this would increase their recall, while possibly reducing their precision.

As can be seen in Table 3 and Supplementary Table 5, `CAMMiQ` achieved the best recall and F1 score (see Supplementary Note 7 for a definition), and the second best precision for the 11 `species-level-bacteria` query sets (the three queries with uneven coverage were excluded). Its precision and recall were particularly impressive for first 7 challenging queries (labeled with prefix "Least"), where `CAMMiQ` was an order of magnitude better than the alternatives in terms of both measures. On these queries, tools other than `CAMMiQ` assigned only a small proportion of reads to genomes at the species level. This is because none of them employ doubly-unique substrings to differentiate species in the index dataset from the same genus. The only exception is Centrifuge, which achieved the best classification precision and the second best recall. For example, on the 3 hypothetically error-free queries (labels ending with **-1** in Table 3), where Centrifuge (in addition to `CAMMiQ` and CLARK) achieved 100% precision. However, Centrifuge's classification performance deteriorated when genomes in queries were likely not present in the corresponding index dataset (Table 2, panels A and B). Note that in principle Kraken2, KrakenUniq and Centrifuge could assign reads to the correct taxonomy higher than the species level. However, as mentioned above, only reads that were assigned to the correct species were considered to be true positives for this benchmark.

## Precision and recall in genome identification on IMMSA and CAMI queries

On the CAMI and IMMSA queries, `CAMMiQ` correctly identified more genomes than the alternative tools with the same abundance cutoff of 0.01% (we consider a genome to have been identified by a tool only if the tool reports its abundance to be ≥0.01% of the total abundance of all genomes), resulting in superior recall values for genome identification (Table 2, panel C; note that recall in genome identification represents the fraction of correctly identified genomes among all genomes in a query set). This is primarily due to `CAMMiQ`'s use of doubly-unique substrings in its query type $\mathcal{A}_2$. Compared to its recall performance, `CAMMiQ` achieved even better precision figures than the alternatives (Table 2, panel D; note that *precision* in genome identification represents the fraction of correctly identified genomes among the set of genomes identified by a given tool), due to fewer false positive identifications. The fact that `CAMMiQ` particularly performs best with respect to the precision values indicates that genomes not present in the index dataset would have the least impact on `CAMMiQ` in comparison to other tools. Note that by postprocessing the output of Kraken2, Bracken manages to improve on the number of identified genomes, and achieves comparable figures to `CAMMiQ`. However, it does not reduce the large number of false positive genome identifications produced by Kraken2, when unknown genomes or genomes outside the index dataset present in the query sample.

## Genome identification and quantification performance on `species-level-bacteria` queries

Next, we evaluated the number of correctly identified genomes by each tool (specific to MetaPhlAn2, the genus corresponding to each genome), as well as the L1 and L2 distances between the true abundance profile and the predicted abundance profile, on the 14 queries involving our `species-level-bacteria` dataset, including the 3 queries with GC bias. As can be seen in Table 3, panels C–E, and Supplementary Table 5, `CAMMiQ` clearly offered the best performance in both identification and quantification. It correctly identified all genomes present in each one of the 14 queries and was not impacted by uneven read coverage or the genome we added to the query Random-20-lognormal-a.g. which was not indexed. Importantly, `CAMMiQ` consistently returned very few false positive genomes for the most challenging queries, and at most one false positive genome for the remaining 4 queries.

Compared to `CAMMiQ`, other tools reported larger number of false negatives in these 14 queries (again we consider a genome to be a "negative", if its reported abundance level is ≤0.01% of the total abundance of all genomes), in particular in the 10 challenging queries (labeled with the prefix "Least") with minimal unique substrings (i.e., *L*-mers). Among them, CLARK and Centrifuge offered the best false negative performance, especially on error free queries. As can be expected, MetaPhlAn2 had the worst performance with respect to false negatives, very likely due to the incompleteness of its marker gene list (we used the latest set of marker genes mpa_v20_m200 in MetaPhlAn2). This also led to a relatively larger L1/L2 distances than the other tools, even for the remaining 4 (easier) queries. Kraken2 and KrakenUniq were also prone to having false negatives, though fewer than MetaPhlAn2. Bracken, in general, could correctly identify a few more genomes than Kraken2, and this improvement in its identification performance also leads to better quantification results (see below).

`CAMMiQ` performs even better with respect to the number of false positive genomes, as demonstrated by its F1 score distribution (see Supplementary Table 5). The alternative tools all returned a large number of false positives in `species-level-bacteria` queries, especially in the first 10 challenging queries, even though all reads in these queries were sampled from (some genome in) the index dataset (see Table 3, panel C). Among them, Centrifuge and Bracken usually performed better on the 10 challenging queries with fewer 'unique' genomes; while KrakenUniq and CLARK performed better on the remaining 4 (easier) queries. Kraken2 showed the worst performance with respect to the false positives: it outputs more than a third of the genomes from the index dataset even for the three error free queries. In many of the datasets, these false positives were eliminated by Bracken's postprocessing of Kraken2's output; unfortunately, in other query datasets, e.g., Random-20-uniform and Random-100-uniform, Bracken introduced additional false positives. MetaPhlAn2 identified only limited number of genomes (and few true positive genomes) in all queries in general, so it had a comparable performance to `CAMMiQ` with respect to false positives. However, its F1 scores were not as good as `CAMMiQ`'s (see Supplementary Table 5).

Note that `CAMMiQ` not only correctly identified all genomes, but also predicted their abundances reasonably close to the true values. As can be seen in Table 3, `CAMMiQ` outperformed all other tools on both L1 and L2 errors, typically offering a factor of 3 - 4× improvement over the second best alternative. Interestingly, even when the coverage across each genome were non-uniform, `CAMMiQ`'s $\mathcal{A}_3$ type of query was only mildly impacted. As noted earlier, on the 10 challenging queries (especially those with sequencing errors), all alternative tools except MetaPhlAn2 output hundreds of false positive genomes. As a consequence, their predictions for the abundances of the true positive genomes were smaller than the true abundance values. This is particularly the case for Kraken2 and KrakenUniq: even though they identified the majority of the true positive genomes correctly, their

reported abundance values were all close to 0; this results in their L1 distances to be very close to 1.

### Evaluation of computational resources on species level queries

We compared the running time and memory usage of CAMMiQ, Kraken2/Bracken, KrakenUniq, CLARK, MetaPhlAn2, and Centrifuge in building the index and responding to the queries; see Table 4. As can be seen CAMMiQ performs better than all alternatives in running time - including those tools that aim to index *all* unique *k*-mers (KrakenUniq and CLARK), and *all* substrings (Centrifuge) - with respect to both query time and index construction time. The only exception is Kraken2 (MetaPhlAn2 uses a pre-built index and so it can not be compared against others with respect to index construction time), however, Kraken2's overall accuracy is worse than the others across the species-level queries. Since MetaPhlAn2 uses a pre-built index (See Supplementary Note 6) it avoids the expensive index construction process. This, however, results in many false negatives (See Subsections Genome identification and quantification performance on species-level-bacteriaqueries and Performance of CAMMiQ at the strain level). CAMMiQ also supports pre-built indices. Compared to the other tools and methods, the sizes of these pre-built indices are much smaller (Table 4, Panel B), due to the sparsification of unique and doubly unique substrings, allowing convenient transfer and fast downloading. Note that we do not report the time for loading the index into memory for any of the tools, since this is performed only once.

All of our experiments were run on a Linux server equipped with 40 Intel Xeon E7-8891 2.80 GHz processors, with 2.5 TB of physical memory and 30 TB of disk space. The ILP solver used by CAMMiQ in the initial implementation is IBM ILOG CPLEX 12.9.0. We have also ported the code to use the ILP solver Gurobi 9.1.0.

### Assessing the use of variable-length and doubly-unique substrings in species-level-bacteria queries

Due to its unique algorithmic features CAMMiQ outperforms available alternatives on the CAMI, IMMSA and the species-level-bacteria query sets. A key question is: what is the maximum potential improvement in performance one can expect through the use of (i) variable-length substrings as opposed to fix length *k*-mers, and (ii) doubly-unique substrings in addition to unique substrings? Here, we evaluate both of these algorithmic features in the context of the species-level-bacteria dataset we constructed (see Table 1). For that, we compare the proportion of *L*-mers (for read length $L = 100$) from each genome $s_i$ in our species-level-bacteria index dataset that are unique or doubly-unique (and thus is utilized by CAMMiQ) with the proportion of *L*-mers that include a unique *k*-mer (and thus can be utilized by CLARK and others) for $k = 30$.

Figure 1a summarizes our findings: on the horizontal axis, the genomes are sorted with respect to the proportion of unique and doubly-unique *L*-mers they have; the vertical axis depicts this proportionality (from 0.0 to 1.0). The figure shows the proportion of unique *L*-mers, doubly-unique *L*-mers, the combination of unique and doubly-unique *L*-mers (all utilized by CAMMiQ), as well as the *L*-mers that include a unique *k*-mer (utilized by, e.g., CLARK) for each genome depicted on the horizontal axis. As can be seen, roughly three quarters of all genomes in this dataset are easily distinguishable since a large fraction of their *L*-mers include a unique *k*-mer. However, about a quarter of the genomes in this dataset can benefit from the consideration of doubly-unique substrings, especially when their abundances are low. In particular, 66 of these 4122 genomes/species have extremely low proportions (each ≤1%) of unique 100-mers. At the extreme, the species *Francisella sp. MA06-7296* does not have a single unique 100-mer and the species *Rhizobium sp. N6212* does not have any 100-mer that includes a unique 30-mer (in fact any substring of length $\leq L_{max} = 50$). These two species cannot be identified by, e.g., CLARK in any microbial mixture, regardless of their abundance values.

Figure 1b depicts the inverse proportionality of doubly-unique *L*-mers in comparison to unique *L*-mers among 50 genomes that have the lowest proportion of unique *L*-mers - for $L = 100$. The inverse-proportionality of unique or doubly-unique *L*-mers for a genome corresponds to the number of reads to be sampled (on average) from that genome to guarantee that the sample includes one read that would be assigned to the correct genome. In the absence of read errors, this guarantees correct identification of the corresponding genome in the query. Note that, in half of these 50 genomes, almost all *L*-mers are doubly-unique. This implies that any query involving one or more of these genomes could only be resolved by CAMMiQ and no other tool.

We further assessed whether the usage of unique and doubly-unique substrings can lead to robust genome identification and quantification performance in practice, by evaluating the distribution of these substrings across the genome. In principle, the more evenly these substrings are distributed across a genome, the less likely CAMMiQ's quantification performance can be impacted by queries composed of genomes with small alterations to the corresponding index genomes. As can be seen in Fig. 1c, d, unique and doubly unique substrings span the entire genome on most of the species in our species-level-bacteria index dataset, not significantly biased towards any functionally annotated region by NCBI (i.e., gene, CDS, ncRNA, rRNA, tRNA, tmRNA or plasmid). Even when the numbers of unique or doubly-unique substrings are relatively small in a genome (for example, the last 3 genomes in Fig. 1d), they are still well distributed, helping CAMMiQ with that genome's identification as well as quantification. We would like to note here that even though some genomes have very few unique substrings, implying that they would be difficult to identify through the use of alternative methods, because of their (well distributed) doubly-unique substrings, CAMMiQ can identify and quantify them accurately. Consider, for example, the last genome in Fig. 1d, *Rhizobium sp. N1341* in which the only unique substrings are located on the plasmids. However, since there are sufficiently many doubly-unique substrings on the chromosome, this species could still be identified and quantified by CAMMiQ, through the $\mathcal{A}_2$ or $\mathcal{A}_3$ type of query.

### Performance of CAMMiQ at the strain level

In the next experiment, we evaluated CAMMiQ's performance (with default parameters) on queries composed from our strain-level dataset that consists of 614 Human Gut related genomes of bacterial strains from 409 species[59] as described in Supplementary Note 5.2. As can be seen in Table 5, CAMMiQ managed to identify and accurately quantify all strains in the queries HumanGut-random-100-1 and HumanGut-random-100-2, and > 96% strains in the other two queries, with almost no false positives. Other tools benchmarked against CAMMiQ lead to either more false negative (KrakenUniq, CLARK, MetaPhlAn2) genomes, or more false positive identifications (Kraken2, Centrifuge). Furthermore, their quantification performance (Table 5, panel B) is worse than CAMMiQ.

### Performance of CAMMiQ on real single-cell metatranscriptomic queries

Our final set of experiments involve "real" metatranscriptomic reads from human monocyte-derived dendritic cells (moDCs)[60]. Because CAMNMiQ's most powerful type $\mathcal{A}_3$ query is not suitable for RNA-seq data (due to high variance in read coverage), we employed $\mathcal{A}_1$ and $\mathcal{A}_2$ queries. We remind the reader that $\mathcal{A}_1$ only uses unique substrings in query reads and returns the genomes in the index for which there is at least one such substring. On the other hand, $\mathcal{A}_2$ computes the smallest set of genomes in the index that include all unique or doubly-unique substrings across the query reads.

Each query was composed of all high quality, non-human scRNA-seq reads from the corresponding single cell[60]. For guaranteeing this, we filtered out all scRNA-seq reads which (i) possibly originate from the

**Table 4 | Comparison of the running times required by** `CAMMiQ`, **Kraken2, KrakenUniq, CLARK Centrifuge, Bracken, and MetaPhlAn2**

| Performance measure | Query set/ Index dataset | CAMMiQ | Kraken2/Bracken | KrakenUniq | CLARK | Centrifuge | MetaPhlAn2 |
|---|---|---|---|---|---|---|---|
| A. Index Construction | `species-level-all` | 14435s | 7826s | 24225s | 54602s | 17755s | N/A |
| Time/Memory | | 1147.0G | 22.2G | 242.3G | 303.2G | 257.3G | |
| | `species-level-bacteria` | 7794s | 4623s | 13304s | 35413s | 17216s | |
| | | 715.9G | 20.0G | 167.6G | 234.6G | 257.3G | |
| | `strain-level` | 688s | 451s | 1575s | 3111s | 1275s | |
| | | 83.9G | 20.0G | 37.2G | 67.1G | 22.3G | |
| B. Index Size On Disk | `species-level-all` | 14.4G | 5.1G | 226.3G | 106.4G | 11.7G | 1.1G |
| | `species-level-bacteria` | 8.9G | 11.2G | 155.2G | 71.8G | 7.3G | |
| | `strain-level` | 0.7G | 1.3G | 21.3G | 7.5G | 0.2G | |
| C. Query Time/Memory | IMMSA-buccal-12 | 14.2s | 7.7s | 19.9s | 29.5s | 59.3s | N/A |
| | | 165.8G | 5.1G | 226.8G | 103.0G | 11.7G | |
| | IMMSA-citypark-48 | 41.6s | 25.8s | 73.9s | 77.6s | 87.6s | |
| | | 165.8G | 5.2G | 226.9G | 103.1G | 11.7G | |
| | IMMSA-gut-20 | 14.3s | 11.0s | 23.7s | 37.5s | 54.0s | |
| | | 165.8G | 5.1G | 226.9G | 103.0G | 11.7G | |
| | IMMSA-house-30 | 23.3s | 15.1s | 42.0s | 47.0s | 63.9s | |
| | | 165.8G | 5.2G | 226.9G | 103.0G | 11.7G | |
| | IMMSA-house-20 | 17.7s | 10.9s | 25.5s | 31.9s | 54.7s | |
| | | 165.8G | 5.2G | 226.8G | 103.0G | 11.7G | |
| | IMMSA-soil-50 | 84.1s | 37.8s | 153.2s | 136.3s | 138.6s | |
| | | 165.8G | 5.2G | 226.9G | 103.5G | 11.7G | |
| | IMMSA-simBA-525 | 128.8s | 84.6s | 265.2s | 210.6s | 218.3s | |
| | | 173.5G | 5.3G | 227.0G | 104.3G | 11.7G | |
| | IMMSA-nycsm-20 | 18.6s | 9.1s | 25.1s | 32.5s | 55.8s | |
| | | 165.8G | 5.2G | 226.9G | 103.0G | 11.7G | |
| | CAMI-LC-1 | 3982.0s | 1565.8s | 7282.9s | 8765.8s | 7218.3s | |
| | | 168.9G | 7.0G | 227.9G | 135.4G | 11.9G | |
| | CAMI-MC-1 | 4013.3s | 1555.4s | 8596.0s | 8158.0s | 6277.2s | |
| | | 168.9G | 7.0G | 233.5G | 135.4G | 11.8G | |
| | CAMI-MC-2 | 3920.5s | 1550.5s | 8436.4s | 8418.9s | 6011.4s | |
| | | 168.9G | 7.2G | 232.7G | 135.3G | 11.8G | |
| | CAMI-HC-1 | 3906.4s | 1557.0s | 9762.5s | 8020.8s | 6022.0s | |
| | | 168.9G | 7.0G | 252.8G | 135.3G | 11.8G | |
| | CAMI-HC-2 | 3866.8s | 1561.3s | 9605.8s | 8131.5s | 6145.6s | |
| | | 168.9G | 7.0G | 252.3G | 135.3G | 11.8G | |
| | CAMI-HC-3 | 3893.6s | 1683.8s | 9934.8s | 8472.9s | 6082.5s | |
| | | 168.9G | 7.0G | 255.8G | 135.3G | 11.8G | |
| | CAMI-HC-4 | 3934.1s | 1631.3s | 9941.5s | 7872.3s | 6028.6s | |
| | | 168.9G | 7.0G | 255.1G | 135.3G | 11.8G | |
| | CAMI-HC-5 | 3941.2s | 1553.0s | 9649.6s | 8460.1s | 6501.9s | |
| | | 168.9G | 7.0G | 253.1G | 135.3G | 11.8G | |
| | Least-20-uniform-2 | 151.7s | 52.2s | 286.5s | 402.5s | 309.2s | 608.8s |
| | | 94.6G | 11.4G | 67.9G | 81.3G | 7.3G | 1.3G |
| | Least-quantifiable-20-uniform-2 | 180.5s | 50.8s | 284.1s | 416.0s | 317.8s | 569.6s |
| | | 93.3G | 11.4G | 77.1G | 81.3G | 7.3G | 1.3G |
| | Least-20-genera-uniform-2 | 143.5s | 53.0s | 273.4s | 285.2s | 242.3s | 438.9s |
| | | 92.6G | 11.4G | 81.4G | 81.0G | 7.3G | 1.3G |
| | Least-20-genera-lognormal | 153.6s | 61.8s | 268.8s | 289.4s | 253.9s | 442.7s |
| | | 93.3G | 11.4G | 77.8G | 81.0G | 7.3G | 1.3G |
| | Random-20-uniform | 198.4s | 48.1s | 284.0s | 315.1s | 207.7s | 516.7s |
| | | 95.4G | 11.4G | 92.7G | 81.3G | 7.3G | 1.3G |
| | Random-20-lognormal | 167.8s | 48.8s | 294.5s | 367.6s | 226.8s | 569.3s |
| | | 95.7G | 11.4G | 89.3G | 81.3G | 7.3G | 1.3G |
| | Random-20-lognormal-a.g. | 49.8s | 14.7s | 78.9s | 75.2s | 58.1s | 188.9s |
| | | 94.3G | 11.3G | 51.9G | 80.3G | 7.3G | 1.3G |
| | Random-100-uniform | 906.0s | 215.4s | 1261.5s | 1536.2s | 969.3s | 2401.4s |
| | | 111.9G | 11.9G | 143.3G | 85.7G | 7.4G | 1.4G |

Index construction time/memory: time and peak RAM usage required by each tool to build the index upon the `species-level-all`, `species-level-bacteria` and `strain-level` index dataset, with 32 threads, with an exception that CLARK only supports a single thread to built the index; MetaPhlAn2 used a collection of pre-built marker genes, which may require several days to complete; for Kraken2 we included the time to preprocess the input genomes (fasta files). Index size on disk: the total index size on disk for the three index datasets required by each software tool. Query time/ memory: time and peak RAM usage to assign all reads in each query set to one (or two) genome(s) when running with a single thread; for `CAMMiQ` we included the required ILP running time and memory usage for $\mathcal{A}_2$ type query on CAMI and IMMSA benchmark query sets and $\mathcal{A}_3$ type query on `species-level-bacteria` query sets; for all tools except MetaPhlAn2, time to load the index into main memory were excluded; for MetaPhlAn2 we measured the total alignment time. Note that Bracken index construction and queries will post-process Kraken2 output, so we group them together. For all memory measurements 1G = $2^{30}$ Bytes.

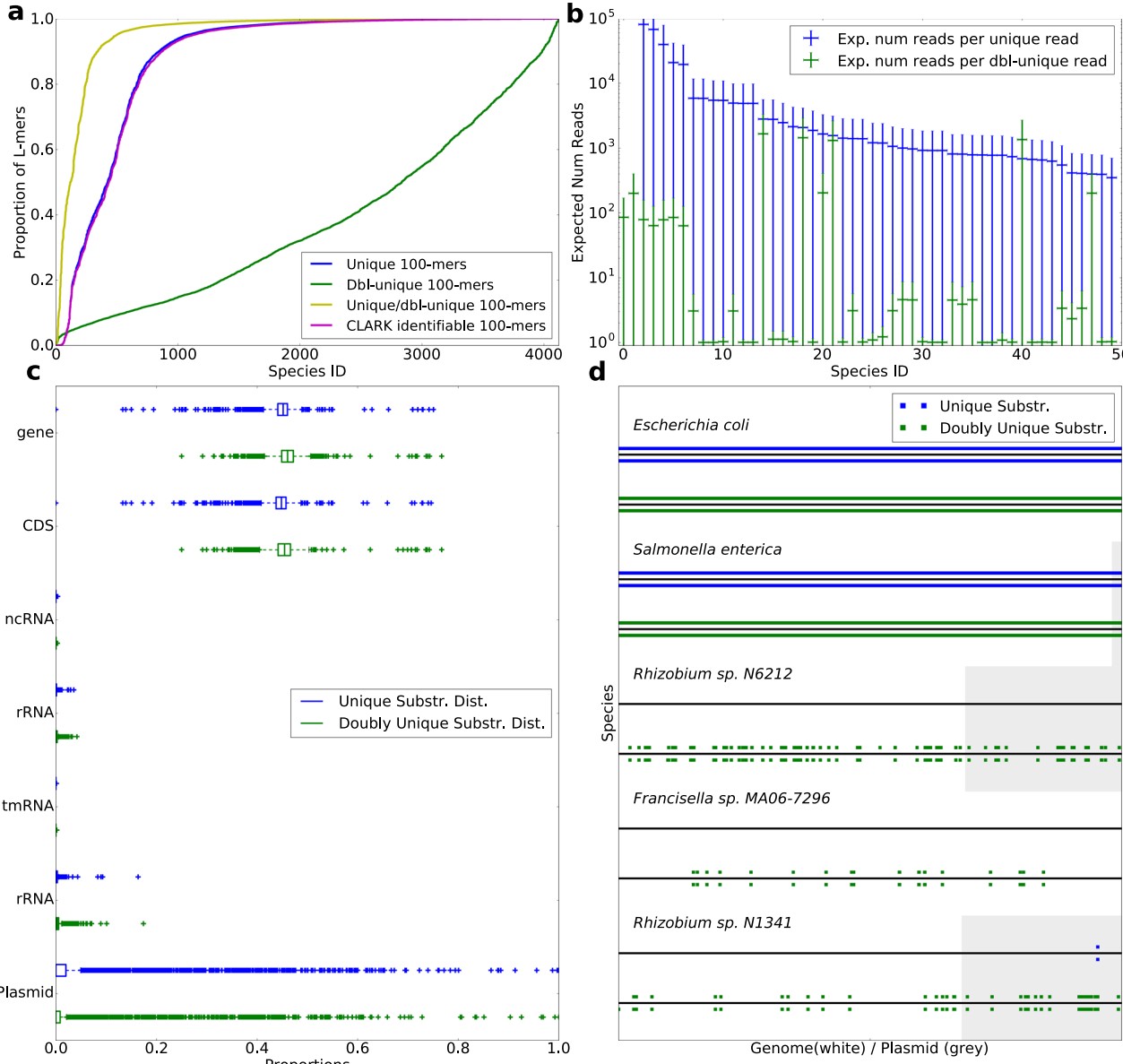

**Fig. 1 | The advantage of using variable-length and doubly-unique substrings.**
**a** The proportion of $L$-mers (for $L = 100$) that include a unique substring (in blue), a doubly-unique substring (in green), or either a unique or a doubly-unique substring (in yellow) in the `species-level-bacteria` dataset of 4122 genomes from NCBI RefSeq; these $L$-mers, when presented as reads are utilizable by `CAMMiQ` (when $L_{max} = 100$). Also included are the proportion of $L$-mers that include a unique $k$-mer ($k = 30$) and thus are utilized by CLARK (in purple). For each plot, the genomes are independently sorted with respect to the corresponding proportions in ascending order. **b** The expected number of reads (of length $L = 100$) needed to capture one read containing a unique substring (in blue) as well as one read containing a doubly-unique substring (in green) in the 50 genomes with the lowest proportion of unique $L$-mers. Also included are the range for each value within the corresponding standard deviation. For the analyses visualized in this figure, $L_{max} = 50$. **c** The distribution of unique (in blue) and doubly-unique (in green) substrings across different functional annotations on the 4122 genomes in `species-level-bacteria` index dataset. Each boxplot shows the median and interquartiles, with whiskers extending to the most extreme data points within 1.5*interquartile range. **d** The loci of unique (in blue) and doubly unique (in green) substrings on 2 representative genomes (E. coli, and S. enterica), and 3 genomes with the least number of unique $L$-mers. Plasmids are marked with gray. The unique/doubly unique substrings on forward strand are plotted above each black line; the unique/doubly unique substrings on reverse strand are plotted below each black line.

human genome, or (ii) have low sequence quality and "complexity", or (iii) map to 16S or 23S ribosomal RNAs on the two *Salmonella* genomes (to avoid incorrect assignment of reads due to "barcode hopping").

Following the original study[60], we categorized each cell into one of the 5 groups: infected cells that were confirmed to contain (1) STM-LT2 or (2) STM-D23580 strain of intracellular *Salmonella*; bystander cells that were exposed to (3) STM-LT2 or (4) STM-D23580 strains, but confirmed to not contain intracellular *Salmonella*; and (5) cells that were mock-infected and sequenced as controls. For each query, we compared the number of reads `CAMMiQ` assigned uniquely to STM-LT2

or STM-D23580 genomes against those aligned and assigned either by the GATK PathSeq[10] tool or blastn[11] (see Supplementary Note 9).

Figure 2 summarizes our results on this data set. In Fig. 2a, we demonstrate that compared to the GATK PathSeq approach, `CAMMiQ`'s $\mathcal{A}_1$ type queries were more sensitive with respect to read assignment. On average, `CAMMiQ` identified (roughly) an order of magnitude more unique STM-LT2 or STM-D23580 reads in each cell, demonstrating its potential to better identify intracellular organisms at subspecies or strain level. Note that `CAMMiQ`'s performance is comparable or slightly better than that of blastn. However `CAMMiQ` is several orders of

magnitude faster than blastn or GATK PathSeq. (CAMMiQ only took a total of 65.3s for computing $\mathcal{A}_1$ type queries and an additional 2.5s for computing $\mathcal{A}_2$ type queries on the entire query set, outperforming GATK PathSeq, which required 29628.1s, or blastn, which is typically slower).

The abundances reported by each of the three tools (measured by unique read counts) of *Salmonella* were substantially higher in the infected cells compared to the mock-infected controls. More importantly, cells known to be infected with or exposed to a particular strain indeed include significantly more reads from that strain. Interestingly, CAMMiQ as well as blastn reported that cells infected with or exposed to a particular strain also contain reads unique to the other strain. This is possibly due to sequencing errors or incorrect cell assignments for these reads.

In Fig. 2b, we compare CAMMiQ's $\mathcal{A}_2$ type queries with its $\mathcal{A}_1$ type queries (as well as GATK PathSeq and blastn) with respect to the number of cells they correctly identify to include STM-LT2 or STM-D23580 strains. For that we vary the minimum number of reads that need to be identified by each tool to report a given strain, and for each such value we indicate how many cells are reported to include the STM-LT2 strain (on the vertical axis) vs the STM-D23580 strain (on the horizontal axis). With the exception of the third subpanel a method with a plot closer to the diagonal is less sensitive. As can be seen CAMMiQ's $\mathcal{A}_2$ type queries are more sensitive than not only its $\mathcal{A}_1$ type queries but also GATK PathSeq and blastn. However, they also introduce some potential false positive calls (e.g., in the third subpanel panel corresponding to the controls). This could be due to additional reads utilized by $\mathcal{A}_2$ queries impacted by read errors or incorrect assignments of these reads to cells.

## Discussion

We have introduced CAMMiQ, a new computational tool to identify microbes in an HTS sample and to estimate abundance of each species or strain. CAMMiQ is based on a principled approach that starts by defining formally the following algorithmic problem that has not been fully addressed by any available method. Given a set $\mathcal{S}$ of distinct genomic sequences of any taxonomic rank, build a data structure so as to identify and quantify genomes in any query, composed of a mixture of reads from a subset of $\mathcal{S}$. CAMMiQ is particularly designed to handle genomes that lack unique features; for that, it reduces the aforementioned identification and quantification problems to a combinatorial optimization problem that assigns substrings with limited ambiguity (i.e., doubly-unique substrings) to genomes so that, in its most general $\mathcal{A}_3$ type query, each genome is "uniformly covered". Uniform coverage is a simplifying assumption we employ in our theoretical analysis since which genomes are represented in a query are not known in advance. In practice, the coverage for genomic sequences might be biased by GC content[61,62]. We do not employ this assumption in CAMMiQ implementation for $\mathcal{A}_1$ and $\mathcal{A}_2$ type queries, which are more suitable for transcriptomic sequences. Our experiments on the *Salmonella* scRNAseq dataset indeed show that CAMMiQ delivers good results on scRNAseq queries work well even though the reads are skewed by variable expression and the selection biases of single-cell technology. Because each such substring has limited ambiguity, the resulting combinatorial optimization problem can be efficiently solved through the existing integer program solvers IBM CPLEX and Gurobi.

One potential limitation of CAMMiQ is that it relies on a database of reference genomes. In the context of medical microbiology this is a reasonable assumption since virtually all clinically-relevant microbes detected in new patients are known and have some similar genome sequenced and in RefSeq. The reliance on a reference database is more problematic in the context of studying environmental samples, in which new and rare taxa might be found by methods that do not rely on reference genomes. Our results on the CAMI benchmark data set provide reassurance that CAMMiQ performs well even when many

genomes and plasmids are absent from the reference database. Another potential limitation is that the memory required by CAMMiQ index construction is relatively high. However, CAMMiQ supports pre-built indices on commonly used databases for metagenomic studies, e.g., (the latest version of) the RefSeq bacteria, viruses and archaea database. Compared to the other tools and methods, the sizes of these pre-built indices are much smaller, due to the sparsification of unique and doubly unique substrings, allowing convenient transfer and fast downloading. The prebuilt CAMMiQ index for all index datasets are available via the GitHub link provided in the Code Availability statement. In addition, as shown for the experiments summarized in Table 4, the memory requirements for CAMMiQ queries are comparable to those of other widely used packages and within the capabilities of currently available computers.

Provided that the doubly-unique substrings of a given genome are not all shared with one other genome, the use of doubly-unique substrings increases CAMMiQ's ability to identify and quantify this genome within a query. In case the dataset to be indexed involves several genomes with high levels of similarity, CAMMiQ's data structure and its combinatorial optimization formulation could be generalized to include "triply" or "quadruply" unique substrings, but this is not yet implemented. In summary, using principled methods from combinatorial optimization and string algorithms, CAMMiQ delivers better sensitivity and specificity than widely-used existing methods on practical genome classification and quantification methods.

## Methods

The input to CAMMiQ is a set of *m* genomes $\mathcal{S} = \{s_i\}_{i=1}^m$, possibly but not necessarily all from the same taxonomic level (each genome here may be associated with a genus, species, subspecies, or strain), to be indexed. Although we describe CAMMiQ for the case where each $s_i \in \mathcal{S}$ is a single string, we do not assume that the genomes are fully assembled into a single contig. The string representing a genome could simply be a concatenation of all contigs from genome $s_i$ and their reverse complements, with a special symbol $\$_i$ between consecutive contigs. We call $\mathcal{S}$ the input *database* or synonymously *index dataset*, and we call $i \in \{1, \cdots, m\}$ the *genome ID* of string $s_i$.

A *query* or *query set* for CAMMiQ contains a set of reads $\mathcal{Q} = \{r_j\}_{j=1}^n$ representing a metagenomic mixture. For simplicity, we describe CAMMiQ for reads of homogeneous length *L*; however, our data structure can handle reads of varying length. Given $\mathcal{Q}$, the goal of CAMMiQ is to identify a set of genomes $\mathcal{A} = \{s_1, \cdots, s_a\} \subset \mathcal{S}$ and their respective abundances $p_1, \cdots, p_a$ that "best explain" $\mathcal{Q}$. This is achieved by assigning (selected) reads $r_j$ to genomes $s_i$ such that the implied coverage of each genome $s_i \in \mathcal{A}$ is (roughly) uniform across $s_i$, with $p_i$ as the mean.

CAMMiQ's index data structure involves the collection of shortest *unique* substrings and shortest *doubly-unique* substrings on each genome $s_i$ in $\mathcal{S}$. We call a substring of $s_i$ unique if it does not occur on any other genome $s_j \neq s_i$ in $\mathcal{S}$; a shortest unique substring is a unique substring that does not include another unique substring. Similarly, we call a substring of $s_i$ doubly-unique if it occurs on exactly one other genome $s_j \neq s_i \in \mathcal{S}$; a shortest doubly-unique substring is a doubly-unique substring that does not include another doubly-unique substring. See Supplementary Note 1 for a formal definition for the uniqueness of a substring and Supplementary Fig. 1 for a graphical illustration. CAMMiQ does not maintain the entire collection of shortest unique and doubly-unique substrings of genomes in $\mathcal{S}$; instead, its index contains only a sparsified set of shortest unique and doubly-unique substrings of each $s_i \in \mathcal{S}$ so that no unique and doubly-unique substring is in close proximity (i.e., within a read length) of another in $s_i$. See Section CAMMiQ Index and Supplementary Note 2 for how exactly CAMMiQ sparsifies the collection of shortest unique and doubly-unique substrings.

**Table 5 | CAMMiQ's strain level performance compared to Kraken2, KrakenUniq, CLARK, Centrifuge, and MetaPhlAn2, on the four strain-level queries**

| Performance measure | Query set | CAMMiQ | Kraken2 | KrakenUniq | CLARK | Centrifuge | MetaPhlAn2 |
|---|---|---|---|---|---|---|---|
| A. Num. Correctly | HumanGut-least-25 | 24/26 | 23/167 | 25/40 | 25/40 | 25/49 | ≥18/19 |
| Identified Strains | HumanGut-random-100-1 | 100/101 | 94/241 | 99/101 | 99/99 | 99/112 | ≥74/83 |
| | HumanGut-random-100-2 | 100/102 | 93/197 | 97/98 | 98/98 | 99/121 | ≥67/72 |
| | HumanGut-all | 404/407 | 380/391 | 395/397 | 396/396 | 392/396 | ≥279/305 |
| B. L1 Err. | HumanGut-least-25 | 0.1130 | 0.5026 | 0.4882 | 0.8607 | 0.5195 | 0.2910 |
| | HumanGut-random-100-1 | 0.0209 | 0.2642 | 0.2244 | 0.3677 | 0.2817 | 0.3260 |
| | HumanGut-random-100-2 | 0.0256 | 0.2300 | 0.2058 | 0.3387 | 0.2821 | 0.5066 |
| | HumanGut-all | 0.0517 | 0.2841 | 0.2426 | 0.4004 | 0.2811 | 0.4439 |
| C. L2 Err. | HumanGut-least-25 | 0.0443 | 0.1606 | 0.1597 | 0.1987 | 0.1246 | 0.0737 |
| | HumanGut-random-100-1 | 0.0036 | 0.0476 | 0.0438 | 0.0460 | 0.0420 | 0.0441 |
| | HumanGut-random-100-2 | 0.0044 | 0.0469 | 0.0446 | 0.0461 | 0.0413 | 0.0648 |
| | HumanGut-all | 0.0062 | 0.0255 | 0.0239 | 0.0253 | 0.0201 | 0.0299 |

The number of strains in each query is indicated in the query label, with the exception of the HumanGut-all query, which includes one representative strain from each of the 409 strains in the index. Number of identified strains: the number of true positive strains/the total number of strains identified. L1 (or L2) error: the L1 (or L2) distance between the true relative abundance values and the predicted abundance values, across all strains in the query.

With the (sparsified) collection of shortest unique and doubly-unique substrings, CAMMiQ is sufficiently powerful to answer the following three types of queries. The simplest type of query only involves unique substrings: given a query set $\mathcal{Q}$, it asks for the set of genomes $\mathcal{A}_1 \subseteq \mathcal{S}$ so that each includes at least one (shortest) unique substring that also occur in some read $r_j$ in the query $\mathcal{Q}$. The second, more general query type involves both unique and doubly-unique substrings. It asks to compute $\mathcal{A}_2 \subseteq \mathcal{S}$, the smallest subset of genomes in $\mathcal{S}$ which include all (shortest) unique and doubly-unique substrings that also occur in some read $r_j \in \mathcal{Q}$. Finally, the third and the most general type of query asks to compute the smallest subset $\mathcal{A}_3$ of $\mathcal{S}$ which again include all (shortest) unique and doubly-unique substrings that also occur in some read $r_j \in \mathcal{Q}$, with the additional constraint that the "coverage" of these substrings in each genome $s_i \in \mathcal{A}_3$ is roughly uniform. In addition to the set of genomes $\mathcal{A}_3$, the query also asks to compute the relative abundance of each genome $s_i$ in $\mathcal{A}_3$.

CAMMiQ with its ability to efficiently answer all three queries described above has several advantages over existing methods that rely on fixed-length unique substrings (i.e., unique $k$-mers). (i) Notice that the shorter a unique substring is the more likely it will be sampled (i.e., present in a read sampled from the relevant genome). This is because a substring of length $L' < L$ is included in $L - L' + 1$ potential reads of length $L$ that could be sampled from a genome. Unfortunately, the shorter a substring is, the less likely that it is unique or doubly-unique. A method that uses fixed length $k$-mers needs to have a compromise between the number of unique substrings and the likelihood of sampling each. CAMMiQ gets around this limitation by utilizing unique substrings of any length. CAMMiQ features a lower bound $L_{min}$ and upper bound $L_{max}$ on the lengths of unique and doubly-unique substrings as explained below. (ii) Unique substrings are relatively rare, at least for certain genomes and taxa, but substrings that appear in many genomes provide very limited information about the composition of a query $\mathcal{Q}$. By involving doubly-unique substrings in a query $\mathcal{Q}$, the subset of genomes that could be identified through query $\mathcal{A}_2$ would be larger and more accurate than those that could be identified through query $\mathcal{A}_1$, especially in the extreme case where $\mathcal{Q}$ includes highly similar genomes that do not include any unique substring. (iii) Finally, by introducing the "uniform coverage" constraint, CAMMiQ's $\mathcal{A}_3$ type of

query can identify more accurately the genome(s) where a doubly-unique substring originates. This is because a query of type $\mathcal{A}_2$ may result in significant differences in coverage between unique and doubly-unique substrings of a given genome.

As mentioned above, CAMMiQ builds an index for the sparsified sets of shortest unique and doubly-unique substrings to compute efficiently the sets $\mathcal{A}_1$, $\mathcal{A}_2$ and $\mathcal{A}_3$. For all three query types, CAMMiQ first identifies for each read $r_j$ all unique and doubly-unique substrings it includes; it then assigns $r_j$ to the one or two genomes from which these substrings can originate. To compute $\mathcal{A}_1$, CAMMiQ can simply return the collection of genomes receiving at least one read assignment. To compute $\mathcal{A}_2$, CAMMiQ needs to solve instances of the NP-hard set cover problem, or more precisely, its dual, the hitting set problem where genomes form the universe of items, and indexed strings that appear in query reads form the sets of items to be hit. Even though this is a restricted version of the hitting set problem where each set to be hit contains at most two items, it is still NP-hard due to a reduction to the vertex cover problem. To compute $\mathcal{A}_3$ CAMMiQ solves the combinatorial optimization problem that asks to minimize the variance among the number of reads assigned to each indexed substring of each genome - the solution indicates the set of genomes in $\mathcal{A}_3$ along with their respective abundances.

Details on the composition as well as the construction process for CAMMiQ's index are discussed in Section CAMMiQ Index, as well as Supplementary Notes 1 and 2. The two stages in query processing of CAMMiQ are discussed in Subsections Query processing stage 1: Preprocessing the Reads and Queryprocessing stage 2: ILP formulation. The first stage assigns reads to specific genomes, which is sufficient for computing sets $\mathcal{A}_1$ and $\mathcal{A}_2$. See Section Query processing stage 1: Preprocessing the Reads and Supplementary Note 3 for the criteria we use to assign a read to a genome, based on the indexed substrings that the read includes. The second stage introduces the combinatorial optimization formulation to compute $\mathcal{A}_3$ as a response to the most general query type. See Section Query processing stage 2:ILP formulation for details.

## CAMMiQ Index

To respond to all three types of queries described above, CAMMiQ identifies all unique and doubly-unique substrings of the genomes in $\mathcal{S}$

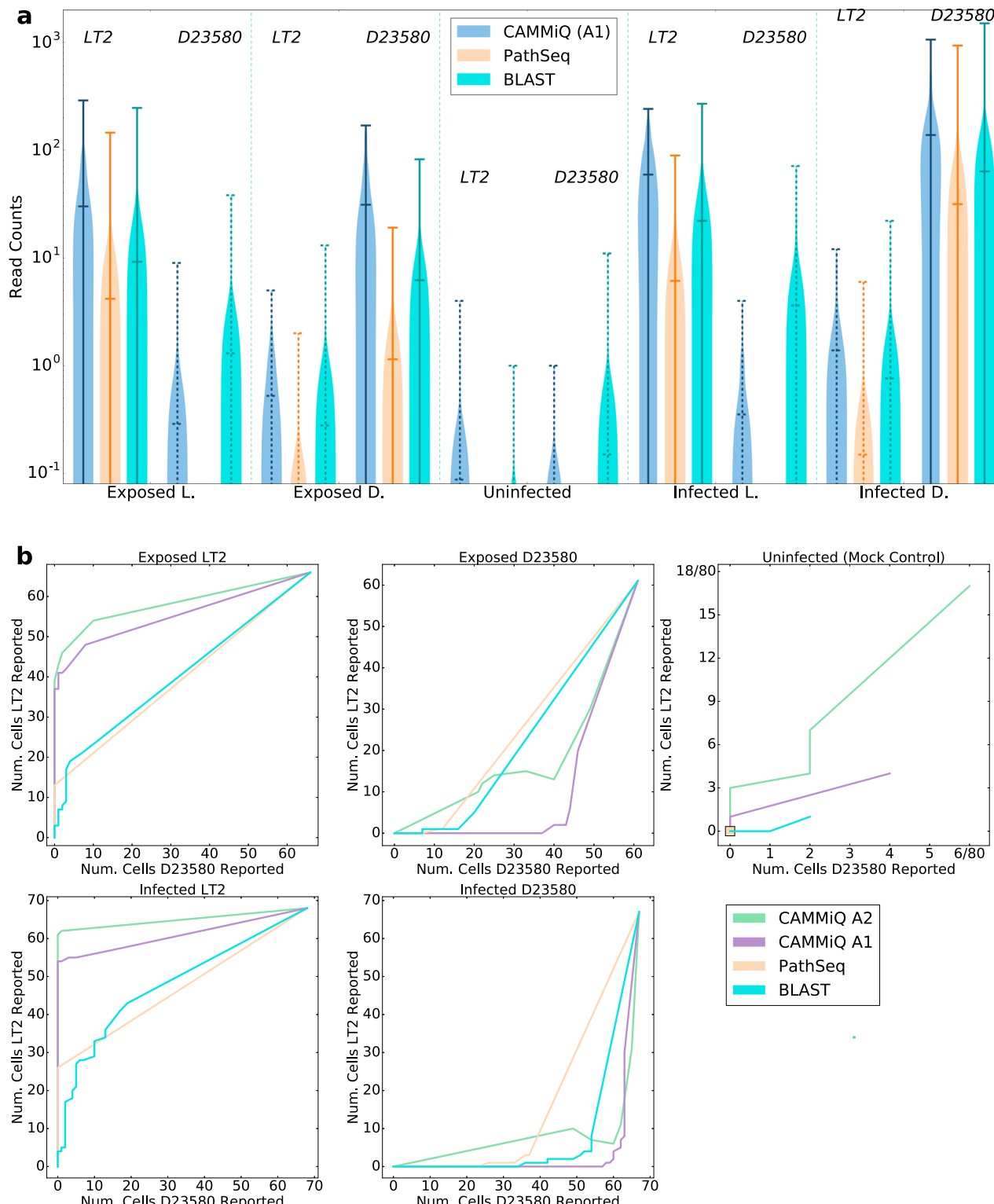

and organizes them in a simple but efficient data structure. Specifically, `CAMMiQ` computes the complete set of shortest unique substrings, $\mathcal{U} = \cup_{i=1}^{m} \mathcal{U}_i$, and the set of shortest doubly-unique substrings, $\mathcal{D} = \cup_{i=1}^{m} \mathcal{D}_i$, where $\mathcal{U}_i$ and $\mathcal{D}_i$ respectively denote the complete set of shortest unique and doubly-unique substrings from genome $s_i$, whose lengths are within the range $[L_{min}, L_{max} \leq L]$. See Supplementary Note 1 for a linear time algorithm to build both $\mathcal{U}$ and $\mathcal{D}$. `CAMMiQ` then sparsifies $\mathcal{U}$ and $\mathcal{D}$ by selecting only one representative substring among those that are in close proximity in each genome, and discarding the

rest; this sparsification step is described in detail in below and the Supplementary Note 2. Finally it builds a collection of tries (trees where the root node represents a substring of length $L_{min}$ and every other internal node represents a single character) to compactly represent and efficiently search for substrings in $\mathcal{U}$ and $\mathcal{D}$.

**Determining $L_{max}$ and $L_{min}$.** In general, as the value of $L_{max}$ increases, so do the numbers of unique and doubly-unique substrings to be considered by `CAMMiQ` - potentially increasing its sensitivity. However query

**Fig. 2 | CAMMiQ performance on the filtered-scRNA-seq queries. a** Comparison of metatranscriptomic reads uniquely assigned to the STM-LT2 strain (the left side in each panel) or STM-D23580 strain (the right side in each panel) by CAMMiQ (blue), GATK PathSeq (orange), and blastn (cyan) across (i) the 66 cells that were exposed to STM-LT2; (ii) the 61 cells exposed to STM-D23580; (iii) the 80 mock-infected cells as controls; (iv) the 68 cells infected with STM-LT2; and (v) the 67 cells infected with STM-D23580. Vertical bars indicate the average and maximum number of reads assigned uniquely to the corresponding strains. Solid lines indicate the strain to which the cells have been exposed or infected with. Ideally the strain with which the cells have been exposed to or infected with should have higher read count values. **b** The number of cells with more than a threshold of $t$ reads uniquely assigned to the STM-LT2 strain (vertical-axis in each panel) or STM-D23580 strain (horizontal-axis in each panel) by CAMMiQ with query type $\mathcal{A}_1$ (purple), GATK PathSeq (orange) and blastn (cyan), for varying values of $t$. We also included the results of CAMMiQ with query type $\mathcal{A}_2$ (green) that uses doubly-unique substrings in addition to unique substrings and thus is not represented in panel (**a**) of the figure. When the threshold $t$ is very high, neither strain could be detected by any method in any of the cells - this corresponds to position $(0, 0)$ in each plot. When the threshold is very low (as low as 0), then both strains can be detected in all cells by all tools - this corresponds to position $(c, c)$ where $c$ corresponds to the total number of cells in a given plot. The plot for an ideal tool should deviate from the diagonal as much as possible - the only exception is the third subpanel which depicts the mock control. The plots for CAMMiQ queries are closer to this ideal than GATK PathSeq or blastn.

type $\mathcal{A}_3$ relies on the read coverage for each unique and doubly-unique substring of each genome; the higher the coverage the better. The read coverage for a unique substring of length $\lfloor L - L/\Delta \rfloor$, for some constant $\Delta > 1$, would roughly be $1/\Delta$-th of the read coverage (of a single nucleotide) of the respective genome. The best tradeoff between these two objectives, i.e., substring length, $\sim (1 - 1/\Delta)$ and coverage, $\sim 1/\Delta$, can be achieved by maximizing their product, i.e., $(1 - 1/\Delta)/\Delta$, which is achieved at $\Delta = 2$. This suggests to choose $L_{\max} = L/2$.

A shortest unique substring $u$, by definition, differs from (at least) one other substring $u'$ by just one nucleotide. The shorter $u$ gets, the more likely a read error impacting $u'$ would modify it to $u$, leading to false positives. We have experimentally observed that unique substrings of length $\leq 25$ could lead to false positives that impact the performance of CAMMiQ; as a consequence, we set the default value of $L_{\min}$ to 26.

**Sparsifying unique substrings.** Let $\mathcal{U}_i$ be the collection of all unique substrings on genome $s_i$. To reduce the index size, CAMMiQ aims to compute a subset $\mathcal{U}_i'$ of $\mathcal{U}_i$, consisting of the minimum number of shortest unique substrings such that every unique substring of length $L$ (i.e., unique $L$-mer) on $s_i$ includes one substring from $\mathcal{U}_i'$. Independently, CAMMiQ also aims to compute a subset $\mathcal{D}_i'$ of $\mathcal{D}_i$, consisting of the minimum number of shortest doubly-unique substrings such that every doubly-unique substring of length $L$ (i.e., doubly-unique $L$-mer) on $s_i$ includes one substring from $\mathcal{D}_i'$. This is all done by greedily maintaining only the rightmost shortest unique or doubly-unique substring in a sliding window of length $L$ on a genome in $\mathcal{S}$. In the remainder of the paper, we denote the number of unique substrings in subset $\mathcal{U}_i'$ by $nu_i$ $(= |\mathcal{U}_i'|)$ and the number of doubly-unique substrings in subset $\mathcal{D}_i'$ by $nd_i$ $(= |\mathcal{D}_i'|)$; we denote the number of unique $L$-mers on $s_i$ by $nu_i^L$ and respectively the number of doubly-unique $L$-mers on $s_i$ by $nd_i^L$. As we prove in Supplementary Note 2, the greedy strategy we employ can indeed obtain the minimum number of shortest unique substrings to cover each unique $L$-mer, provided that each substring in $\mathcal{U}_i$ occurs only once in $s_i$.

**Index organization.** We demonstrate the index structure and query processing for the set of unique substrings $\mathcal{U}$; the processing for doubly-unique substrings is essentially identical to that for unique substrings. Let $h = \min_{u_i \in \mathcal{U}} |u_i|$ be the minimum length of all shortest unique substrings ($h$ is automatically set to $L_{\min}$ if the minimum length constraint is imposed). CAMMiQ maintains a hash table that maps a distinct $h$-mer $w$ to a bucket containing all unique substrings $u_i$ that have $w$ as a prefix. Within each bucket, the remaining suffices of all unique substrings $u_i$, i.e., $u_i[h+1:|u_i|]$, are maintained in a trie (rooted at $u_i[1:h]$) so that (i) each internal node represents a single character; and (ii) each leaf represents the corresponding genome ID. For each read $r_j$ in the query, CAMMiQ considers each substring of length $h$ and its reverse complement and computes its hash value in time linear with $L$ through Karp–Rabin fingerprinting[63]. If the substring has a match in the hash table, then CAMMiQ tries to extend the match until a matching unique substring is found, or until an extension by one character leads to no match. See Fig. 3 for a schematic of the index structure. See Subsection Query processing stage 1: Preprocessing the Reads below for the use of unique and doubly-unique substrings identified for each read to answer the query.

**Query processing stage 1: preprocessing the reads**
Given the index structure on the sparsified set of shortest unique and doubly-unique substrings of genomes in $\mathcal{S}$, we handle each query $\mathcal{Q}$ in two stages. The first stage counts the number of reads that include each unique and doubly-unique substring with the following provision. We call two or more (unique or doubly-unique) substrings in a read "conflict-free" if there is at least one genome that includes all of these substrings. See Supplementary Note 3 for a detailed discussion on conflicting substrings; the conflicts arise due to either sequencing errors or the query including genomes that are not in the database and thus should be avoided. Reads that include more than one unique or doubly-unique substring that is conflict-free contribute to the counting process; all other reads are discarded.

We denote by $c(u_i)$, the counter for the conflict-free reads that include the unique substring $u_i$ and by $c(d_i)$ that for the doubly-unique substring $d_i$. These counters are sufficient to compute the set $\mathcal{A}_1$ as well as $\mathcal{A}_3$, the answer to our most general query type. For computing $\mathcal{A}_2$, CAMMiQ additionally maintains a counter $d(s_k, s_{k'})$ for each pair of genomes $s_k, s_{k'}$, indicating the number of reads in $\mathcal{Q}$ that can originate both from $s_k$ and $s_{k'}$ (i.e., the case (e - iii) in the procedure described in Supplementary Note 3).

The first stage thus produces two count vectors $\mathbf{c}_i^u = (c(u_{i,1}), \cdots, c(u_{i,nu_i}))$ and $\mathbf{c}_i^d = (c(d_{i,1}), \cdots, c(d_{i,nd_i}))$ that indicate the number of (conflict-free) reads that include each unique and doubly-unique substring on each genome $s_i$. Using these vectors, CAMMiQ answers the first type of query by computing $\mathcal{A}_1 = \{s_i : \sum_{l=1}^{nu_i} c(u_{i,l}) > 0\}$. Additionally, through the use of the counters $d(s_k, s_{k'})$, CAMMiQ answers the second type of query by computing $\mathcal{A}_2 = \arg\min |\mathcal{A}' \subset \mathcal{S}|$ such that (i) $s_i \in \mathcal{A}'$ if $\sum_{l=1}^{nu_i} c(u_{i,l}) > 0$ and (ii) $\exists s_i \in \mathcal{A}'$, if $d(s_k, s_{k'}) > 0$ then either $i = k$ or $i = k'$. This is basically the solution to the hitting set problem we mentioned earlier, whose formulation as an integer linear program (ILP) is well known[64]. The genomes returned in $\mathcal{A}_1$ are ranked in decreasing order by the aggregated counter values on unique substrings (i.e., $|\mathbf{c}_i^u|$); and the genomes returned in $\mathcal{A}_2$ are ranked by the aggregated counter values on unique substriungs plus the counter values on doubly-unique substrings (i.e., $|\mathbf{c}_i^u| + |\mathbf{c}_i^d|$).

From this point on, our main focus will be how CAMMiQ answers the third type of query by computing $\mathcal{A}_3$ through an ILP formulation described below.

**Query processing stage 2: ILP formulation**
In its second stage, CAMMiQ computes the list of genomes in the query as well as their abundances through an ILP. Let $\delta_i = 0/1$ be the indicator for the absence or presence of the genome $s_i$ in $\mathcal{Q}$. The ILP formulation assigns a value to each $\delta_i$ and also computes for each $s_i$ its abundance $p_i$, upper bounded by $p_{\max}$ - a user-defined maximum abundance with a default setting of 100, which is introduced to avoid potential

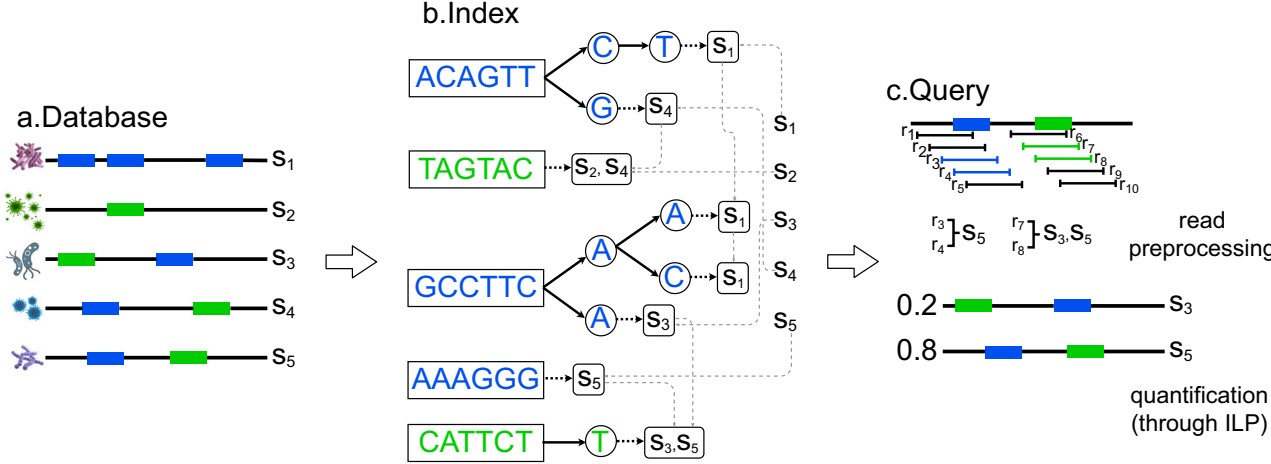

**Fig. 3 | Overview of** `CAMMiQ`**'s index structure.** Strings in blue are unique substrings and those in green are doubly-unique substrings.

anomalies due to sequence contamination.

$$\textbf{Minimize} \quad \sum_i \left( \frac{1}{nu_i} \sum_{l=1}^{nu_i} |c(u_{i,l}) - e(u_{i,l})| + \frac{1}{nd_i} \sum_{l=1}^{nd_i} |c(d_{i,l}) - e(d_{i,l})| \right)$$

$$\textbf{s.t.} \quad e(u_{i,l}) = (L - |u_{i,l}| + 1) \cdot p_i \cdot \frac{1}{L} \cdot (1 - \hat{err})^{|u_{i,l}|} \quad \forall i, l, \text{s.t.}\ 1 \le l \le nu_i \tag{1}$$

$$e(d_{i,l}) = (L - |d_{i,l}| + 1) \cdot (p_i + p_j) \cdot \frac{1}{L} \cdot (1 - \hat{err})^{|d_{i,l}|} \quad \forall i, l\, \text{s.t.}\ 1 \le l \le nd_i \tag{2}$$

$$p_i \le \delta_i \cdot p_{\max} \quad \forall i \tag{3}$$

$$\delta_i = 0 \quad \forall i, \text{s.t.}\ s_i \in M(\mathcal{Q}) \tag{4}$$

$$p_i \ge \delta_i \cdot \min\{L \sum_{l=1}^{nu_i} c(u_{i,l}) \cdot \frac{1}{nu_i^t}, L \sum_{l=1}^{nd_i} c(d_{i,l}) \cdot \frac{1}{nd_i^L}\} \cdot (1 - \epsilon) \quad \forall i, \text{s.t.}\ s_i \notin M(\mathcal{Q}) \tag{5}$$

$$\sum_i |s_i| \cdot p_i \le n \cdot L \tag{6}$$

The objective of the ILP is to minimize the sum of absolute differences between the expected and the actual number of reads to cover a unique or doubly-unique substring. Since each genome may have different numbers of unique and doubly-unique substrings, the sums of differences are normalized w.r.t. $nu_i$ or $nd_i$.

Constraint (1) defines the expected number of reads to cover a particular unique substring $u_{i,l}$, given abundance $p_i$ of the corresponding genome $s_i$. Similarly, constraint (2) defines the expected number of reads to cover a particular doubly-unique substring $d_{i,l}$; in this constraint, $p_i$ and $p_j$ denote the respective abundances of the two genomes $s_i$ and $s_j$ that include (the doubly unique substring) $d_{i,l}$. Specifically the expected coverage of $u_{i,l}$ is $\frac{L-|u_{i,l}|+1}{L} \cdot p_i$ and the expected coverage of $d_{i,l}$ is $\frac{L-|d_{i,l}|+1}{L} \cdot (p_i + p_j)$, provided that the coverage is uniform across a given genome and there are no read errors. To account for read errors, we normalize these coverage estimates respectively by $(1 - \hat{err})^{|u_{i,l}|}$ and $(1 - \hat{err})^{|d_{i,l}|}$; these values represent the probability that, a substring $u_{i,l}$ or $d_{i,l}$ would be error free within a read that has been subject to uniform i.i.d. substitution errors. Here $\hat{err}$ denotes the estimated substitution error rate per nucleotide; and $|w|$ denotes the length of a substring $w$. `CAMMiQ` formulation also allows updates to the expected coverage

according to any given unique or doubly-unique substring's sequence composition (e.g., GC content) to address sequencing biases.

Constraint (3) ensures that the abundance $p_i$ of a genome is 0 if $\delta_i = 0$. Constraint (4) ensures that the solution to the above ILP excludes those genomes whose counters for unique and doubly-unique substrings add up to a value below a threshold - so as to reduce the size of the solution space. More specifically, given a threshold value $\alpha$ ($\alpha$ is introduced to avoid potential false positives due to read errors and genomes that are not in the database; its default value is 0.0001), the constraint excludes those genomes $s_i$ that are in the set of genomes $M(\mathcal{Q})$ whose counters for its unique substrings add up to a value below $\alpha \cdot nu_i^L$, and doubly-unique substrings add up to a value less than $\alpha \cdot nd_i^L$. Formally, $M(\mathcal{Q}) = \{s_i \in \mathcal{S} \mid \sum_{l=1}^{nu_i} c(u_{i,l}) < \alpha \cdot nu_i^L\} \cap \{s_i \in \mathcal{S} \mid \sum_{l=1}^{nd_i} c(d_{i,l}) < \alpha \cdot nd_i^L\}$. Constraint (5) enforces a lower bound on the coverage of each genome $s_i$ in the solution to the above ILP (namely, with $\delta_i = 1$), which must match the coverage $(L \cdot \sum_{l=1}^{nu_i} c(u_{i,l}) \cdot \frac{1}{nu_i^L}$ and $\sum_{l=1}^{nd_i} c(d_{i,l}) \cdot \frac{1}{nd_i^L})$ resulting from the number of reads in $\mathcal{Q}$ that include a unique and doubly-unique substring respectively, i.e., it must be at least $(1 - \epsilon)$ times the smaller one above for a user defined $\epsilon$. Constraint (6) enforces an upper bound on the coverage of each genome $s_i$ in the solution to the above ILP, through making the sum over each $s_i$ of the number of reads produced on $s_i$ based on $p_i$ not exceed the total number of reads $n$. Collectively, the last two constraints ensure that the abundance $p_i$ computed from the ILP matches what is (i.e., the coverage based on read counts) given by $\mathcal{Q}$. As written above, the formulation does not strictly conform to the rules for ILPs because of the use of the absolute value function. We use a standard technique to replace the absolute values in the objective by introducing a new variable $\gamma(u_{i,l}) \ge \max\{c(u_{i,l}) - e(u_{i,l}), e(u_{i,l}) - c(u_{i,l})\}$.

**When to use unique substrings—the error free case**
We now provide a set of sufficient conditions to guarantee the approximate performance that can be obtained with high probability in metagenomic identification and quantification by the use of unique substrings only. These conditions apply to `CAMMiQ` when $c = 1$, as well as CLARK, KrakenUniq, and other similar approaches. In case these conditions are not met, it is advisable to use `CAMMiQ` with $c \ge 2$.

Suppose that we are given a query $\mathcal{Q}$ composed of $n$ error-free reads of length $L$, sampled independently and uniformly at random from a collection of genomes $\mathcal{A} = \{s_1, \cdots, s_a\}$ according to their abundances $p_1, \cdots, p_a$. More specifically, suppose that our goal is to answer query $\mathcal{Q}$ by computing $\mathcal{A}_1$, along with an estimate for the abundance value $p_i$ for each $s_i \in \mathcal{A}_1$, calculated as the weighted number of reads

assigned to $s_i$ according to the procedure described in Section Query processing stage 1: Preprocessing the Reads. Then, the L1 distance between the true abundance values and this estimate will not exceed a value determined by $n$ (number of reads), $a$, and $q_{min}$, the minimum normalized proportion of unique $L$-mers among these genomes. For a given failure probability $\zeta$ and an upper bound on L1 distance $\epsilon$, this translates into sufficient conditions on the values of $n$, $a$ and $q_{min}$ to ensure acceptable performance by the computational method in use.

**Theorem 1.** Let $\mathcal{Q} = \{r_1, \cdots, r_n\}$ be a set of $n$ error-free reads of length $L$, each sampled independently and uniformly at random from all positions on a genome $s_i \in \mathcal{A} = \{s_1, \cdots, s_a\}$, where $s_1, \cdots, s_a$ is distributed according to their abundances $p_1, \cdots, p_a > 0$. Let $p_i' = \frac{p_i \cdot n_i^L}{\sum_{i'=1}^{a} p_{i'} \cdot n_{i'}^L}$ be the corresponding "unnormalized" abundance of $p_i$ for $i = 1, \cdots, a$, where $n_i^L$ denotes the total number of $L$-mers on $s_i$. Let $q_1, \cdots, q_a > 0$ be the proportion of unique $L$-mers on $s_1, \cdots, s_a$ respectively; $p_{min} = \min\{p_i\}_{i=1}^{a}$; $q_{min} = \min\{q_i\}_{i=1}^{a}$. Then,

- (i) With probability at least $1 - \zeta$, each $s_i$ can be identified through querying $\mathcal{Q}$ if $n \geq \frac{2(a+1) + \ln(1/\zeta)}{(p_{min} q_{min})^2}$

- (ii) With probability at least $1 - \zeta$, the L1 distance between the predicted abundances $\hat{p}_1, \cdots, \hat{p}_a$ by setting $\hat{p}_i = \frac{c_i/q_i}{n}$ and the true (unnormalized) abundances $p_1', \cdots, p_a'$ is at most $\epsilon$ if $n \geq \frac{2(a+1) + \ln(1/\zeta)}{(\epsilon q_{min})^2}$.

- (iii) Given $n$ such reads in a query, with probability at least $1 - \zeta$, the L1 distance between the predicted abundances $\hat{p}_1, \cdots, \hat{p}_a$ by setting $\hat{p}_i = \frac{c_i/q_i}{n}$ and the true (unnormalized) abundances $p_1', \cdots, p_a'$ is bounded by $\sqrt{\frac{2[\ln(1/\zeta) + (a+1)]}{n q_{min}^2}}$.

Where $c_i$ denotes the number of reads assigned to $s_i$.
See Supplementary Note 4 for a proof.

## Reporting summary

Further information on research design is available in the Nature Research Reporting Summary linked to this article.

## Data availability

There are four index datasets (`species-level-all`, `species-level-bacteria`, `strain-level` and `subspecies-level`) and associated query sets used in this paper. All of the four index datasets include a subset of all (complete) bacterial, viral and archaeal genomes from NCBI's RefSeq database, which is available at is available at ftp://ftp.ncbi.nlm.nih.gov/genomes/refseq. For the `species-level-all` index dataset, we use the release version 205 of RefSeq, which can be found at https://ftp.ncbi.nlm.nih.gov/refseq/release/release-notes/archive/RefSeq-release205.txt. The complete list of 16418 genomes can be found in https://github.com/algo-cancer/CAMMiQ/blob/master/README.md. The corresponding IMMSA queries can be found privately at http://ftp-private.ncbi.nlm.nih.gov/nist-immsa/IMMSA. A publicly available copy of the above directory is available at https://ftp.ncbi.nlm.nih.gov/pub/catSMA/for_Kaiyuan. The CAMI queries as well as the ground truth files can be found at at http://gigadb.org/dataset/100344. For the `species-level-bacteria` index dataset, we use the release version 93 of RefSeq, which can be found at https://ftp.ncbi.nlm.nih.gov/refseq/release/releasenotes/archive/RefSeq-release93.txt. The complete list of 4122 genomes can be found in https://github.com/algo-cancer/CAMMiQ/blob/master/README.md. The corresponding queries were generated by a python script `CAMMiQ-simulate`, which is available along with the software repo https://github.com/algo-cancer/CAMMiQ; these queries are available upon request. For the `strain-level` index dataset, we use the release version 93 of RefSeq, which can be found at https://ftp.

ncbi.nlm.nih.gov/refseq/release/release-notes/archive/RefSeq-release93.txt. The list of human gut associated bacteria was obtained in the Supplementary Table 1 from https://www.nature.com/articles/s41587-018-0009-7. The complete list of 614 genomes can be found in https://github.com/algo-cancer/CAMMiQ/blob/master/README.md. The corresponding queries were also generated by running `CAMMiQ-simulate`, which is available along with the software repo https://github.com/algo-cancer/CAMMiQ; these queries are available upon request. For our `subspecies-level` index dataset, we use the release version 93 of RefSeq, which can be found at https://ftp.ncbi.nlm.nih.gov/refseq/release/release-notes/archive/RefSeq-release93.txt. The complete list of 3395 genomes can be found in https://github.com/algo-cancer/CAMMiQ/blob/master/README.md. The corresponding scRNA-seq queries can be obtained from https://www.ncbi.nlm.nih.gov/bioproject/PRJNA437328.

## Code availability

The source code of CAMMiQ, under the MIT license, is publicly available at github https://github.com/algo-cancer/CAMMiQ (https://doi.org/10.5281/zenodo.7102588).

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

## Acknowledgements

This research was supported in part by the Intramural Research Program of the National Institutes of Health, National Cancer Institute. This work utilized the computational resources of the NIH HPC Biowulf cluster. (http://hpc.nih.gov). Y.Y. acknowledges support from the NIH grant 5R01AI143254. We thank Dr. Moses Stamboulian for the details of the strain level index dataset.

## Author contributions

K.Z., A.F.E., and S.C.S. initially formulated the problem of metagenomic abundance estimation. K.Z., A.F.E., and S.C.S. developed the index structure and query process. K.Z. implemented the proposed solution. K.Z. performed the comparison with other software tools. Y.Y. and her lab provided the strain-level index dataset. W.R., A.A.S., and E.R. provided the analysis of scRNA-seq dataset. A.A.S provided the support of testing codes on NIH HPC Biowulf cluster. A.A.S. performed the blastn analysis of *Salmonella* strains. K.Z., A.A.S., W.R., J.X., and S.C.S. co-wrote the manuscript. A.A.S. provided further proofreading of the manuscript. E.R., A.F.E., Y.Y., and S.C.S. supervised the study.

## Funding

## Competing interests

The authors declare no competing interests.
