## [Peer Review File · Nature Communications]

Reviewers' comments:

Reviewer #1 (Remarks to the Author):

Zhu et al present an original method for detection and quantification of known genomes within short-read metagenomic datasets. Their approach is based on first identifying and indexing variable length unique and doubly-unique substrings within the genome collection; and then reformulating presence detection and abundance estimation problems as ILP on the counts of the indexed substrings.

I will try to focus on major critique and suggestions.

The authors correctly note that they are comparing to the tools that were not exactly designed to solve the problem that they state. But I am very surprised that they do not benchmark against the 'metagenomic strain identification' tools that have a much better fit for their stated goal of detection and abundance estimation on the level of individual genomes from the database. In particular PathoScope 2, Sigma, maybe GSMer, etc. Even GATK PathSeq was used for comparison in only one experiment. Moreover (and very surprisingly) the widely used Bracken tool, which is aimed at estimating species abundances based on Kraken output using expectation maximization algorithm was not evaluated. Neither was Centrifuge, which belongs to the same class of methods (read taxonomic classifiers), but also can benefit from matches of various sizes (rather than having fixed k-mer size) and, as far as I remember, also integrates the EM-based read re-assignment. Interestingly, as far as I could see, both Bracken and Centrifuge were highlighted by the Simon et al 2019 review, which is cited by the authors as a source for picking the tools for benchmarking. Quoting the review: "Notably, Bracken - a post-processing step intended to improve abundance estimates by Kraken - does provide more accurate abundances at the species level. This step is a simple and worthwhile addition to Kraken for better abundance estimates." And again, while tools like PathoScope, Sigma, etc are not covered by the scope of that review they are probably most relevant for the stated problem of strain-level microbial detection and quantification.

In my opinion the biggest problem of the approach though is that in most metagenomic applications it is unreasonable to assume that exact or even very similar (in terms of the whole genome alignment, not ANI) genome will be present in the database. And since the database probably needs to be deduplicated to some reasonable degree, even the presence of reference with fully matching gene content in many cases will be quite unlikely. (That is the reason why many tools for strain-level metagenomic analysis have to operate with accessory genes composition, perform strain typing based on variants within core genes, etc). And it seems that in the current state the method is not designed to handle this case (presence of SNVs and indels; gene loss; horizontal transfer, etc). It is

certainly true for most novel and 'general' type of query (A_3) when the abundances are also being estimated, which the authors do not even benchmark on the IMMSA datasets, where the reads were synthetically generated from a set of genomes, which was different than the references constituting the database. But even species-level detection queries for IMMSA datasets resulted in a very concerning level of false detections, making the practical utility of the current version of the software questionable. At least in the absence of a comparison with other reference-based tools specifically aimed at species identification.

Last but not the least, the study does not present any attempt at the analysis of real metagenomic datasets.

Overall, I like the doubly-unique substrings idea, even though I am still to be convinced that it will make a difference outside of specifically designed challenging datasets. Joint ILP reformulation of the detection and abundance estimation is also interesting, although I'm legitimately worried about its robustness to sequencing biases, growing populations and other sources of coverage variation. But even more importantly, I think any practical method for metagenomic analysis must demonstrate at least some level of robustness to genomic differences, accessory gene loss and horizontal transfers.

Summing up, concerns about analysis of reads originating from a set of genomes not perfectly matching the ones in the database; lack of benchmarking on real metagenomic data; and lack of comparison against most relevant competing approaches leaves me no choice but advice against publication of the study in its current form.

Minor comments

- . Theoretical analysis leading to both L_{\min} and L_{\max} in section "Determining L_{\max} and L_{\min} " is highly controversial.
- . Constraint (5) in ILP formulation is probably missing denominator of the form $(L - |u| + 1)$
- . Please comment if there is an interplay between sparsification of unique and doubly-unique strings or if those are independent. Namely, from the description it seems that sliding L-windows are considered and we ensure that if the window contains doubly-unique strings then at least one of them will be kept. But some of the L-mers themselves could be unique rather than doubly unique, potentially leading to almost the same strings (differing by 1-bp) being present in unique and doubly-unique sparsified indices. Or will doubly-unique sparsification procedure exclude such L-mers from the consideration?

. Referring to values reported by Kraken etc as ‘true abundances’ on page 13 and in supplement section 6.3 is very confusing for the reader. Especially since ‘true abundance’ is also used in the more usual sense of actual underlying values. I understand that the authors tried to distinguish between ‘molecule-level’ vs base ‘pair-level’ abundance values, but please investigate if there is a less ambiguous conventional terminology to do that.

. I could not find the parameters used for various tools in the text or supplement. Was there any threshold used to extract a set of species ‘detected’ by Kraken etc? If not then would using one make sense to achieve more reasonable recall/precision trade-off?

. It was surprising to see so much attention devoted to the analysis of the recall and precision of individual read classification, considering the goals of the study and the types of queries discussed in methods.

. While explaining why KrakenUniq performed the best on Random-20-lognormal-a.g. two different explanations are given in two different places. “...likely CAMMiQ is impacted more due to its substring sparsification” vs KrakenUniq having more false positive hits, bringing overall abundances closer to expected.

. I am confused why high levels of false positive species/genome detections get associated with low ‘recall’ rather than ‘precision’?

. ‘A method that uses fixed length k-mers needs to have a compromise between the number of unique substrings and the likelihood of sampling each.’ -- in the paragraph around that sentence sequencing errors are ignored. Not mentioning discussion of genomic differences, which are ignored throughout the paper :)

. ‘Another novel feature of our data structure is its use of the variable length substrings’ -- please add background on Centrifuge (at least) which uses variable length substrings.

. Mash Screen paper (Ondov et al 2019) reference should probably be added to the list 43-46 (it seems most relevant for this type of metagenomic analysis).

Reviewer #2 (Remarks to the Author):

The authors present CAMMiQ, a variable-length substring approach to metagenomic classification and relative abundance estimation. Variable-length substrings via shortest unique substrings (or doubly unique) is a novel approach to the problem, with very detailed algorithmic contributions in the main text and supplementary materials. CAMMiQ has three query modes, A1, A2, and A3. A1 asks for a set of genomes such that each includes at least one shustring and also occurs in a read. A2 asks for the smallest subset of genomes which include shustrings and doubly unique substrings and

also occur in a read. A3 asks the same as A2, but requires the coverage to be roughly uniform. The authors show highly competitive results to some of the most widely used tools in the application domain, with respect to runtime, recall, and precision, and across a diverse set of simulated datasets and real datasets (scRNA-seq).

Major

1. The authors state in the introduction: "Provided that the sequence data to be analyzed are genomic, the distribution of HTS reads from a given species or strain should be roughly uniform... In the context of metagenomic abundance estimation, however, this principle is under-utilized.",. Published methods like GRID-MG (<https://www.nature.com/articles/s41467-018-07240-8>) and SMEG (<https://www.science.org/doi/10.1126/sciadv.aaz2299>), in addition to known biases specific to amplification bias (due to low GC/high GC regions, among other factors) stand in direct conflict with these statements (unless the word roughly in roughly uniform is meant to capture this?). Given the objective of the ILP is to identify a set of genomes where the coverage of the near-unique substrings is uniform, additional experimental results highlighting the effect of non-uniform coverage for metagenomic datasets with respect to CAMMiQ is required (A3 query mode). For example, the authors could introduce coverage bias into the species-level-bacteria simulated dataset, specific to the least* datasets, and also run modes A1 and A2 to compare/contrast the performance of all query modes on these simulated datasets.

2. While I fully agree that variable-length substrings are a useful concept in the setting of taxonomic classification, they have been leveraged before in a related setting, or the exact setting of metagenomic classification. Insignia (https://academic.oup.com/nar/article/37/suppl_2/W229/1126701, published in 2009) leveraged maximal unique matches for identifying sequence signatures for diagnostic/identification purposes. Kaiju used maximal exact matches (MEMs) for the task of metagenomic classification (<https://www.nature.com/articles/ncomms11257>), as well as Centrifuge (<https://genome.cshlp.org/content/26/12/1721.short>). Comparisons to both of these methods would be appropriate given their ability to handle variable-length matches.

3. The authors fail to cite prior work that innovated the use of shortest unique substrings (shustrings) in genomic analyses (<https://bmcbioinformatics.biomedcentral.com/articles/10.1186/1471-2105-6-123>). While a different application area (genome analysis and comparison, this work needs to be cited and put into proper context. The manuscript, as is, reads as though shustrings have not previously been used for genomic comparison.

4. It is non-intuitive to understand what effect user-defined parameters (-Lmax and -h) have on the results since a heuristic value is chosen for both. A comparative analysis of datasets used in Table 3 would help readers understand parameter tuning for their datasets.

5. With respect to rigor and reproducibility, I may have missed it but could not find scripts to reproduce the results specific to the experiments presented in the manuscript, nor the raw results from the experiments.

6. How would undercollapsed or overcollapsed contigs for a specific genome (due to copy number variation within a genome and shared repetitive sequence across genomes) in a given assembly affect CAMMiQs performance? Some discussion of CAMMiQ's robustness to misassembly error (when dealing with contigs from assembled metagenomes) is needed.

7. With respect to performance (query time and memory usage), the maximum size dataset that was included was 21.5 million reads. This is roughly the size of a MiSeq run; showing how CAMMiQ scales up to 100 million reads and beyond will help reinforce its computational efficiency/applicability to large metagenomic datasets.

8. With respect to Identification precision and recall, it seems as though MetaPhlan2 would have superior F1 scores, offering up to 5X improvement in recall (IMMSA-gut-20) at the cost of a 1.4X drop in precision, or 4X improvement in recall (IMMSA_buccal-12) at a cost of 1.4X drop in precision. Including an F1-score in addition to just Precision and Recall in all tables makes comparison easier and more straightforward to assess the pros and cons of CAMMiQ in these settings.

9. The authors should also include L2 Error in Table 3 and Table 5.

10. It would be extremely helpful for interpretation of CAMMiQs superior performance via its algorithmic strategy to know where the shortest unique and doubly unique substrings are located in these genomes (including functional annotation). If these are found in mobile genetic elements vs clade-specific marker genes, etc. For well-annotated genomes in the simulated mixtures, this should be reasonably straightforward to add in.

Minor

11. Clarify: “Because any extension of a unique substring is also unique”, I think this should read “Because any extension of a shortest unique substring is also unique”. Maximal unique substrings do not exhibit this property.

12. I strongly recommend including a comparison to MetaPalette, a k-mer painting approach to metagenomic taxonomic profiling, with a sparsity-promoting optimization procedure. Metapalette was shown to have a competitive run time compared to many of the included tools the authors included, and superior profiling performance.

13. Ganon should be mentioned as an alternative approach that performs comparatively well to many of the tools included in this study (<https://www.ncbi.nlm.nih.gov/pmc/articles/PMC7355301/>).

14. Doubly unique is confusing terminology, recommend using alternative phrasing to avoid confusion.

15. The discussion has statements completely encapsulated in parenthesis, unclear why this is done, e.g.: (This is a simplifying assumption we employ since which genomes are represented in a query are not known in advance. In practice, the coverage for genomic sequences might be biased by GC content. We do not employ this assumption for A and A2 type queries which are more suitable for transcriptomic sequences. Our experiments on the Salmonella scRNAseq dataset indeed show that these types of queries work well even though the reads are skewed by variable expression and the selection biases of single-cell technology.)

Reviewers' comments:

Reviewer #1 (Remarks to the Author):

Zhu et al present an original method for detection and quantification of known genomes within short-read metagenomic data sets. Their approach is based on first identifying and indexing variable length unique and doubly-unique substrings within the genome collection; and then reformulating presence detection and abundance estimation problems as ILP on the counts of the indexed substrings.

I will try to focus on major critique and suggestions.

The authors correctly note that they are comparing to the tools that were not exactly designed to solve the problem that they state. But I am very surprised that they do not benchmark against the 'metagenomic strain identification' tools that have a much better fit for their stated goal of detection and abundance estimation on the level of individual genomes from the database. In particular PathoScope 2, Sigma, maybe GSMer, etc. Even GATK PathSeq was used for comparison in only one experiment. Moreover (and very surprisingly) the widely used Bracken tool, which is aimed at estimating species abundances based on Kraken output using expectation maximization algorithm was not evaluated. Neither was Centrifuge, which belongs to the same class of methods (read taxonomic classifiers), but also can benefit from matches of various sizes (rather than having fixed k-mer size) and, as far as I remember, also integrates the EM-based read re-assignment. Interestingly, as far as I could see, both Bracken and Centrifuge were highlighted by the Simon et al 2019 review, which is cited by the authors as a source for picking the tools for benchmarking. Quoting the review: "*Notably, Bracken - a post-processing step intended to improve abundance estimates by Kraken - does provide more accurate abundances at the species level. This step is a simple and worthwhile addition to Kraken for better abundance estimates.*" And again, while tools like PathoScope, Sigma, etc are not covered by the scope of that review they are probably most relevant for the stated problem of strain-level microbial detection and quantification.

Response: We thank the reviewer for the suggestions. Centrifuge and Bracken were added to the list of relevant packages cited in the second paragraph of the Introduction on **page 1**. More importantly, we now performed comparisons of CAMMiQ to Centrifuge and Bracken, which are both alignment-free methods, across all our data sets; please see the newly added columns in **Tables 2, 3, 4, and 5**; the commands and parameters used to run Centrifuge and Bracken for these newly added tests can be found in the new **Supplementary Section 6**. However, we could not run PathoScope 2; it is not well maintained and does not have clear documentation. Additionally, PathoScope 2 as well as Sigma require alignment of each read to the entire reference genome collection. Alignment of reads, especially to an entire collection of reference genomes is a highly time consuming task. As with other alignment-free methods, a major advantage of CAMMiQ is that it does not require read alignment. Therefore, we benchmarked CAMMiQ against similar alignment-free tools for a fair comparison. The reviewer is correct in noting that PathSeq is alignment-based; the reason why we compared CAMMiQ with PathSeq on scRNA-seq data is that the original study from which we obtained the data set used PathSeq;

we aimed to demonstrate that the use of CAMMiQ instead of PathSeq in that study would have saved substantial time while achieving similar (in fact better) cell stratification.

In my opinion the biggest problem of the approach though is that in most metagenomic applications it is unreasonable to assume that exact or even very similar (in terms of the whole genome alignment, not ANI) genome will be present in the database. And since the database probably needs to be deduplicated to some reasonable degree, even the presence of reference with fully matching gene content in many cases will be quite unlikely. (That is the reason why many tools for strain-level metagenomic analysis have to operate with accessory genes composition, perform strain typing based on variants within core genes, etc). And it seems that in the current state the method is not designed to handle this case (presence of SNVs and indels; gene loss; horizontal transfer, etc). It is certainly true for most novel and ‘general’ type of query (A_3) when the abundances are also being estimated, which the authors do not even benchmark on the IMMSA data sets, where the reads were synthetically generated from a set of genomes, which was different than the references constituting the database. But even species-level detection queries for IMMSA data sets resulted in a very concerning level of false detections, making the practical utility of the current version of the software questionable. At least in the absence of a comparison with other reference-based tools specifically aimed at species identification.

Response: We thank the reviewer for this comment. First, we would like to point out that all existing widely-used alignment-free methods assume that an identical or highly similar genome is present in the database. Furthermore, none of the existing alignment-free methods have been explicitly designed to handle the presence of SNVs, indels, gene loss, horizontal transfer (the latter two phenomena typically result in genomic breakpoints). In fact, even the slow alignment-based methods are not designed to handle structural alterations to genomes such as gene loss or horizontal transfer. The key practical benefit of CAMMiQ is that even though it is similar to other alignment-free methods in its underlying premise, CAMMiQ obtains better precision and recall (and especially many fewer false positive detections) without the use of additional computational resources.

Nevertheless, to address the reviewer’s comment directly, we have now benchmarked CAMMiQ against alternative approaches on two additional data sets. The first data set was established by the Critical Assessment of Metagenome Interpretation (CAMI) challenge. The CAMI query data sets we used in our experiments are comprised of (1) a low complexity (in terms of number of species in the mixture) metagenome containing bacteria and viruses (denoted ‘CAMI-LC-1’); (2) two medium complexity metagenomes containing archaea, bacteria and viruses (denoted ‘CAMI-MC-1’ and ‘CAMI-MC-2’); and (3) five high complexity metagenomes containing archaea and bacteria (denoted ‘CAMI-HC-1’ ... ‘CAMI-HC-5’). How we used the CAMI data sets in our experiments is described in a paragraph spanning **pages 8-9** at the beginning of the **Results** Section. A description of the CAMI data sets we used has also been added to **Table 1** and to **Supplementary Table 2**. The performance of CAMMiQ and other tools on the CAMI data sets can be found in the newly added rows in **Table 2** (panels C and D), **Table 4** (panel B) and **Supplementary Table 5**. Note that these results were obtained by querying these data sets

against our species-level-all index. Importantly, only a small proportion of the CAMI query data sets are genomes from the index data set - the majority of the reads in these query data sets represent added plasmids, unknown species or simulated strains “evolved” from known species that are not in the index data set. As such, these metagenomes contain only about 30-40% of genomes from known species in the above-mentioned kingdoms, and even fewer genomes from the index data set. The second data set consists of three queries, each derived from our own species-level-bacteria data set by making the read coverage (across each genome) uneven, denoted ‘Least-20-uniform-3 (uneven)’, ‘Least-quantifiable-20-uniform-3 (uneven)’ and ‘Least-20-genera-uniform-3 (uneven)’. A description of these query data sets has been added to **Supplementary Section 5.4.3, Table 1, Supplementary Table 3 and Supplementary Table 4**. We queried these data sets against our species-level-bacteria index. On both of the new benchmarking data sets CAMMiQ had better performance (with higher precision and recall) in comparison to all alternative methods.

The alternative methods benchmarked against CAMMiQ have been demonstrated to be highly useful with major impact, e.g. in medicine. In fact we compared CAMMiQ against the methods with the highest impact and visibility in the field, namely Kraken (cited 3067 times and 1545 times since 2020) and its follow-up Kraken2 (cited 1280 times and 1272 times since 2020), KrakenUniq (cited 160 times and 133 times since 2020), MetaPhlAn (cited 1273 times and 310 times since 2020), and its followup MetaPhlAn2 (cited 1081 times and 488 times since 2020), Centrifuge (cited 535 times and 292 times since 2020), Bracken (cited 497 times and 423 since 2020), and CLARK (cited 492 times and 195 times since 2020); note that the citation counts are from Google Scholar (obtained on June 15, 2022). These methods were developed by major groups active in metagenomics and provide a good representation of the area. As evidenced by the number of citations only since the start of 2020, these methods have received a lot of attention and are widely used in case studies, including in the recent discovery of specific intracellular bacteria in cancers. (Nejman, D. et al., The human tumor microbiome is composed of tumor type-specific intracellular bacteria. *Science* 368, 973–980 (2020), Poore, G. D. et al., Microbiome analyses of blood and tissues suggest cancer diagnostic approach. *Nature* 579, 567–574 (2020).) Because of its novel algorithmic formulation, CAMMiQ, is more accurate than all these impactful methods, without using more computational resources, and thus has potential to make practical impact.

Last but not the least, the study does not present any attempt at the analysis of real metagenomic data sets.

Response: The newly added CAMI data set collection (please see the newly added rows to **Tables 1, 2 and 4**) was designed as a benchmark set for real metagenomic data sets. Therefore, we have now benchmarked CAMMiQ against the alternatives using various data sets within CAMI. Because CAMI query data sets have a known ground truth, they provide proper means to benchmark these tools - any data set without known ground truth would not be useful for benchmarking precision or recall.

Overall, I like the doubly-unique substrings idea, even though I am still to be convinced that it will make a difference outside of specifically designed challenging data sets. Joint ILP

reformulation of the detection and abundance estimation is also interesting, although I'm legitimately worried about its robustness to sequencing biases, growing populations and other sources of coverage variation. But even more importantly, I think any practical method for metagenomic analysis must demonstrate at least some level of robustness to genomic differences, accessory gene loss and horizontal transfers.

Response: We thank the reviewer for this comment. We emphasize that in medical applications, identifying known strains in metagenomic samples is the critical bioinformatics problem. Additionally, as per our response to one of the reviewer's previous comments, none of the widely-used existing methods, including alignment-based methods are designed to explicitly handle gene loss or horizontal transfer. However, as, e.g., shown on the CAMI data sets, CAMMiQ obtains better precision and recall than all the highly used and impactful alignment-free methods without requiring additional computational resources, demonstrating its practical utility.

Summing up, concerns about analysis of reads originating from a set of genomes not perfectly matching the ones in the database; lack of benchmarking on real metagenomic data; and lack of comparison against most relevant competing approaches leaves me no choice but advice against publication of the study in its current form.

Response: We have newly used the best known, state of the art benchmarking data sets from CAMI and demonstrated that CAMMiQ performs better than all alignment-free alternatives including those newly suggested by the reviewer; please see the newly added rows for this data set in **Table 2**. The methods we benchmarked against CAMMiQ have had major impact and have been developed by some of the most active groups in the field. The demonstration that CAMMiQ performs better than the most impactful tools in the field on the best known, community developed benchmarking data sets should provide sufficient evidence for its practical utility and future use.

Minor comments

Theoretical analysis leading to both L_{\min} and L_{\max} in section "Determining L_{\max} and L_{\min} " is highly controversial.

Response: We have no idea why this analysis is controversial. We would appreciate some additional explanation.

In general, as the value of L_{\max} increases, so does the number of unique (and doubly-unique) substrings to be considered by CAMMiQ, potentially increasing its sensitivity.

However, query type A_3 relies on the read coverage for each unique (and doubly unique) substring of each genome; the higher the coverage the better.

The read coverage for a unique substring of length $L-L/d$, for some constant $d>1$, would (approximately) have $1/d$ fraction of the read coverage (for a single nucleotide) of the respective genome.

The best tradeoff between these two objectives, i.e. substring length, $\sim(1-1/d)$, and coverage, $\sim 1/d$, can be achieved by maximizing their product, i.e., $(1-1/d)/d$, which is achieved at $d=2$.

This suggests picking $L_{\{max\}} = L/2$.

A shortest unique substring u , by definition, differs from (at least) one other substring u' by just one nucleotide. The shorter it gets, the more likely a read error impacting u' would modify it to u , potentially leading to false positives. As mentioned in the paragraph crossing from **page 5 to page 6** “*We have experimentally observed that unique substrings of length <26 do lead to false positives that impact the performance of CAMMiQ; as a consequence, we set the default value of $L_{\{min\}}$ to 26*”.

Constraint (5) in ILP formulation is probably missing denominator of the form $(L - |u| + 1)$

Response: The variable p_i represents the underlying ‘coverage’ or ‘sequencing depth’ for a nucleotide (not a substring) and does not require ‘length normalization’; i.e., constraint (5) in the ILP formulation on **page 7** is correct.

Please comment if there is an interplay between sparsification of unique and doubly-unique strings or if those are independent. Namely, from the description it seems that sliding L -windows are considered and we ensure that if the window contains doubly-unique strings then at least one of them will be kept. But some of the L -mers themselves could be unique rather than doubly unique, potentially leading to almost the same strings (differing by 1-bp) being present in unique and doubly-unique sparsified indices. Or will doubly-unique sparsification procedure exclude such L -mers from the consideration?

Response: The sparsification is done independently. We have now made this clear in the description of the sparsification step in **Section 2.1** of the revised manuscript. As a consequence, it is indeed theoretically possible that a slightly shifted version of a unique substring may end up being chosen as a doubly unique substring. However, in our experience, the vast majority of the doubly unique substrings are not as such; please see the new **Figure 2D** (the bottom three panels) that demonstrates that a genome may have no unique substrings but many doubly unique substrings.

Referring to values reported by Kraken etc as ‘true abundances’ on page 13 and in supplement section 6.3 is very confusing for the reader. Especially since ‘true abundance’ is also used in the more usual sense of actual underlying values. I understand that the authors tried to distinguish between ‘molecule-level’ vs base ‘pair-level’ abundance values, but please investigate if there is a less ambiguous conventional terminology to do that.

Response: Throughout the paper we aimed to be consistent with the notion of “genome length normalized” abundance used by CAMMiQ, MetaPhlan2, Centrifuge and Bracken, which is different from the total number of reads assigned to a genome, used by Kraken, KrakenUniq and CLARK as abundance. For each given tool, we used the notion of abundance employed by the paper introducing that tool. We have now added a new **Supplementary Section 6.1** where we explain these two notions of abundance in detail. Since in the tables we report not the absolute abundance values, but rather abundance values normalized by the total abundance of all genomes

(which vary between 0 and 1), the values we report for different tools are comparable, even though they do not measure the exact same quantity. If the editors think that we need to pick one of these notions as a basis of our comparisons, we could certainly do so.

I could not find the parameters used for various tools in the text or supplement. Was there any threshold used to extract a set of species ‘detected’ by Kraken etc? If not then would using one make sense to achieve more reasonable recall/precision trade-off?

Response: As mentioned in the first paragraph in **page 9**, we have now added the command line parameters used for CAMMiQ, Kraken2, KrakenUniq, CLARK, Centrifuge, Bracken and MetaPhlan2 to the **Supplementary Section 6**.

Note that in order to have a fair comparison for genome identification across all tools, we have now used the same threshold (i.e. 0.01% of the total abundance; i.e., a genome is considered to be a negative if its abundance is <0.01% of the total abundance of all genomes). This is now described in the first and the third paragraphs of **page 14**, as well as the new **Supplementary Section 6.9**. This resulted in (minor) updates in **Table 2, panel C and D** and **Table 3, panel C and D**.

It was surprising to see so much attention devoted to the analysis of the recall and precision of individual read classification, considering the goals of the study and the types of queries discussed in methods.

Response: This is a matter of taste. Some of the tools that we have compared against CAMMiQ, e.g. CLARK or Kraken, do emphasize the recall and precision for individual read classification. We aimed to demonstrate that even for this measure CAMMiQ performs better than its competitors. We would also like to point out that to some extent, the analysis of read classification is connected to the analysis of whether the read data are biased. We have attempted to make this connection by analyzing the uniformity of coverage, e.g., in the two new panels added to **Figure 2 (panels C and D)**. These demonstrate the (balanced) distribution of unique and doubly-unique substrings across a number of genomes.

While explaining why KrakenUniq performed the best on Random-20-lognormal-a.g. two different explanations are given in two different places. "...likely CAMMiQ is impacted more due to its substring sparsification" vs KrakenUniq having more false positive hits, bringing overall abundances closer to expected.

Response: We thank the reviewer for this observation. We have rewritten an entire paragraph on these queries (the last paragraph in **page 12**), especially since the newly benchmarked tool Centrifuge performs better than all others with respect to classification recall.

I am confused why high levels of false positive species/genome detections get associated with low ‘recall’ rather than ‘precision’?

Reaction: We thank the reviewer for this observation. Indeed, there was an inconsistency in the paper. We have corrected it and have now clarified how we have computed recall and precision

across the paper in **pages 11 (Table 2) and 14** (the paragraph with the headline: “*Precision and recall in genome identification on IMMSA and CAMI queries*”).

‘A method that uses fixed length k-mers needs to have a compromise between the number of unique substrings and the likelihood of sampling each.’ -- in the paragraph around that sentence sequencing errors are ignored. Not mentioning discussion of genomic differences, which are ignored throughout the paper :)

Response: We thank the reviewer for this comment. Even though the presence of sequencing errors (which occurs with a frequency of ~0.1% in Illumina sequencing) and SNVs (between index genomes and sequenced genomes) could have an impact in CAMMiQ’s ability to resolve the origin of a read, the same issue is also valid for k-mer based approaches. In fact a sequencing error or SNV can alter up to k-1 distinct (unique) k-mers - if no sparsification among k-mers is performed by the method (such as CLARK); however, that would alter just one unique substring for CAMMiQ (because of its sparsification step). On one hand, the inability for such k-mer based approaches to distinguish the k-mers originating from the same genomic locus vs those originating from different genomic locations is possibly responsible for their weaker genomic abundance estimation in comparison to CAMMiQ. On the other hand, there are some k-mer based approaches such as Kraken2 which do sparsify unique k-mers. However if a sequencing error or SNV impacts a unique substring shorter than k, it will certainly impact the associated unique k-mer. If on the other hand the unique substring is longer than k, there could be no unique k-mers covering the sequencing error or SNV, implying that the impacted read would not be assigned to the correct genome anyway.

Note that CAMMiQ’s ILP formulation additionally provides a corrected estimate on the number of reads covering a unique substring, based on the observed sequencing error rate; see **constraints (1) and (2) in page 7**. Likely due to these features, CAMMiQ performs better than all k-mer based approaches in genomic abundance estimation.

‘Another novel feature of our data structure is its use of the variable length substrings’ -- please add background on Centrifuge (at least) which uses variable length substrings.

Response: We have now mentioned Centrifuge in the Introduction section, specifically in the third paragraph of **page 2**: “*Alignment-free methods typically rely on exact string matching [references to Centrifuge and Kaiju] or k-mer (substrings of length k) “matches” to obtain a taxonomic assignment for every read.*”; we also have newly benchmarked Centrifuge against CAMMiQ on all species and strain level queries - please see the **Tables 2, 3, 4 and 5** for results.

Mash Screen paper (Ondov et al 2019) reference should probably be added to the list 43-46 (it seems most relevant for this type of metagenomic analysis).

Response: The Mash Screen paper has been added to the bibliography and is cited in paragraph 3 on **page 2**.

Reviewer #2 (Remarks to the Author):

The authors present CAMMiQ, a variable-length substring approach to metagenomic classification and relative abundance estimation. Variable-length substrings via shortest unique substrings (or doubly unique) is a novel approach to the problem, with very detailed algorithmic contributions in the main text and supplementary materials. CAMMiQ has three query modes, A1, A2, and A3. A1 asks for a set of genomes such that each includes at least one shustring and also occurs in a read. A2 asks for the smallest subset of genomes which include shustrings and doubly unique substrings and also occur in a read. A3 asks the same as A2, but requires the coverage to be roughly uniform. The authors show highly competitive results to some of the most widely used tools in the application domain, with respect to runtime, recall, and precision, and across a diverse set of simulated data sets and real data sets (scRNA-seq).

Response: We thank the reviewer for the nice summary of our contributions..

Major

1. The authors state in the introduction: “Provided that the sequence data to be analyzed are genomic, the distribution of HTS reads from a given species or strain should be roughly uniform... In the context of metagenomic abundance estimation, however, this principle is under-utilized.”. Published methods like GRID-MG (<https://www.nature.com/articles/s41467-018-07240-8>) and SMEG (<https://www.science.org/doi/10.1126/sciadv.aaz2299>), in addition to known biases specific to amplification bias (due to low GC/high GC regions, among other factors) stand in direct conflict with these statements (unless the word roughly in roughly uniform is meant to capture this?). Given the objective of the ILP is to identify a set of genomes where the coverage of the near-unique substrings is uniform, additional experimental results highlighting the effect of non-uniform coverage for metagenomic data sets with respect to CAMMiQ is required (A3 query mode). For example, the authors could introduce coverage bias into the species-level-bacteria simulated data set, specific to the least* data sets, and also run modes A1 and A2 to compare/contrast the performance of all query modes on these simulated data sets.

Response: The reviewer is correct in the sense that what is meant by “roughly uniform” coverage can easily be corrected for sequence content bias (e.g. amplification bias) in the CAMMiQ formulation. We now cite the two papers mentioned by the reviewer in the first paragraph of the **Discussion section**: “*In practice, the coverage for genomic sequences might be biased by GC content [References for GRID-MG and SMEG].*”.

Additionally, following the suggestion of the reviewer, we have newly compared CAMMiQ against other tools on a new data set, which consists of three queries, each derived from our own species-level-bacteria data set by making the read coverage (on each genome) non-uniform; these are respectively denoted ‘Least-20-uniform-3 (uneven)’, ‘Least-quantifiable-20-uniform-3 (uneven)’ and ‘Least-20-genera-uniform-3 (uneven)’. We queried them against our

species-level-bacteria index. As can be seen in our experimental results in **Table 3**, the non-uniform coverage across each genome had virtually no impact on genome identification (the number of true positive and false positive genomes identified), and negligible impact on abundance estimation (the L1 and newly added L2 error).

2. While I fully agree that variable-length substrings are a useful concept in the setting of taxonomic classification, they have been leveraged before in a related setting, or the exact setting of metagenomic classification. Insignia (https://academic.oup.com/nar/article/37/suppl_2/W229/1126701, published in 2009) leveraged maximal unique matches for identifying sequence signatures for diagnostic/identification purposes. Kaiju used maximal exact matches (MEMs) for the task of metagenomic classification (<https://www.nature.com/articles/ncomms11257>), as well as Centrifuge (<https://genome.cshlp.org/content/26/12/1721.short>). Comparisons to both of these methods would be appropriate given their ability to handle variable-length matches.

Response: We thank the reviewer for pointing this out. We would like to note that we are aware that the use of variable length substrings in string matching predates CAMMiQ and even Insignia (see for example *Efficient approximate and dynamic matching of patterns using a labeling paradigm*, S.C. Sahinalp, U. Vishkin, *IEEE FOCS, 1996*, which used variable length substrings for improving the efficiency of string matching). However Insignia was developed for a very different application than CAMMiQ: it aims to find a “match” for a query sequence (e.g. a read) in a reference genome sequence (or multiple sequences) with a functionality like that of BLAST. Importantly, as mentioned in the original paper, Insignia “*finds a series of overlapping k-mer signatures and reports these longer chains as a single region, where every k-mer in the chain is guaranteed to be unique.*”. As such, Insignia uses an intuitive heuristic for chaining multiple k-mer matches for a longer alignment. In contrast CAMMiQ is the first algorithmic method to identify variable length unique (or doubly-unique) substrings in a collection of genomes that can be used to assign a query read to a particular genome which is provably correct.

The reviewer is correct in pointing out that Centrifuge uses string matching through the use of the Burrows-Wheeler Transform (BWT) of a collection of genomes with the aim of finding the longest substring of a read that exists exactly in these genomes. We have now compared Centrifuge against CAMMiQ across our benchmarking data sets (please see **Tables 2, 3 and 5**). From a conceptual point however, Centrifuge implicitly indexes *all* substrings (shorter than read length) of *all* genomes. On the other hand CAMMiQ’s only indexes shortest unique and doubly-unique substrings of a genome explicitly, in a hash table. In fact, after sparsification, it only indexes those unique and doubly-unique substrings that are non-overlapping. As such, Centrifuge and CAMMiQ offer two different conceptual approaches to read classification. Because of its unique approach, CAMMiQ is not only more accurate but also is faster than Centrifuge, even though CAMMiQ solves an ILP problem after searching for strings in its index.

Kaiju similarly performs read classification using a similar string matching approach employed by Centrifuge. Unfortunately, Kaiju works on protein sequences rather than genomic or

transcriptomic sequences so it can not be compared against CAMMiQ directly. Nevertheless we now cite Kaiju as a method that uses string matching (**page 2, paragraph 3**) for performing metagenomic classification: “*Alignment-free methods typically rely on exact string matching [references to Centrifuge and Kaiju] or k-mer (substrings of length k) “matches” to obtain a taxonomic assignment for every read.*”.

3. The authors fail to cite prior work that innovated the use of shortest unique substrings (shustrings) in genomic analyses (<https://bmcbioinformatics.biomedcentral.com/articles/10.1186/1471-2105-6-123>). While a different application area (genome analysis and comparison, this work needs to be cited and put into proper context. The manuscript, as is, reads as though shustrings have not previously been used for genomic comparison.

Response: We thank the reviewer for bringing this paper to our attention. We now cite the paper in the **Introduction** (top paragraph of **page 3**) and give appropriate credit - “*For $c=1$ such substrings were called “shortest unique substrings” and were utilized in comparing genomes of various species [reference to the above mentioned paper]*”.

4. It is non-intuitive to understand what effect user-defined parameters (-Lmax and -h) have on the results since a heuristic value is chosen for both. A comparative analysis of data sets used in Table 3 would help readers understand parameter tuning for their data sets.

Response: We thank the reviewer for this remark. Note that the parameter -h is specific to the hashing method used (it is the length of the prefix of substrings that is not chained in the hash table). Since it does not impact which substrings are hashed but rather alters how they are hashed, it only impacts speed and not accuracy. In contrast, -Lmax is the maximum length of a substring that is hashed. The reviewer is right in pointing out that, at least in theory, as -Lmax increases, the number of unique substrings in a genome and thus the accuracy of CAMMiQ should improve. In practice, however, CAMMiQ’s performance was minimally impacted after increasing the value of -Lmax, for any of the four index data sets we used. Since these are good representatives of an index data set to be used in practice, we believe setting -Lmax to read length/2 is a good rule of thumb. The explanation about -h and -Lmax has been added in **Supplementary Section 8**. If the editors find it necessary to perform such an experiment we would be happy to add it to the paper.

5. With respect to rigor and reproducibility, I may have missed it but could not find scripts to reproduce the results specific to the experiments presented in the manuscript, nor the raw results from the experiments.

Response: We thank the reviewer for this remark. We have added a new Supplementary Section 6 to the paper that provides the specific commands and parameter settings used for CAMMiQ and other methods we benchmarked in our comparisons.

6. How would undercollapsed or overcollapsed contigs for a specific genome (due to copy number variation within a genome and shared repetitive sequence across genomes) in a given assembly affect CAMMiQs performance? Some discussion of CAMMiQ's robustness to misassembly error (when dealing with contigs from assembled metagenomes) is needed.

Response: We thank the reviewer for this remark. As the reviewer has observed, an incorrectly characterized number of copies of a genomic region will result in uneven coverage of the genome. Since query types A1 and A2 only consider the presence of reads matching a genome, they would not be impacted by uneven coverage. Even for query type A3, uneven coverage of a genome would not have an impact in its identification; however this could have an impact in its abundance estimate. To assess this impact we have performed new experiments involving queries of type A3 with unevenly covered genomes, as we have detailed in our response to the reviewer question 1.

7. With respect to performance (query time and memory usage), the maximum size data set that was included was 21.5 million reads. This is roughly the size of a MiSeq run; showing how CAMMiQ scales up to 100 million reads and beyond will help reinforce its computational efficiency/applicability to large metagenomic data sets.

Response: We thank the reviewer for the suggestion. We have now included eight new queries from CAMI (Critical Assessment of Metagenome Interpretation) challenge, each with ~100 million reads as suggested. As can be seen in **Table 4**, the running time of CAMMiQ increased only linearly with the number of reads in these queries, demonstrating its scalability.

8. With respect to Identification precision and recall, it seems as though MetaPhlan2 would have superior F1 scores, offering up to 5X improvement in recall (IMMSA-gut-20) at the cost of a 1.4X drop in precision, or 4X improvement in recall (IMMSA_buccal-12) at a cost of 1.4X drop in precision. Including an F1-score in addition to just Precision and Recall in all tables makes comparison easier and more straightforward to assess the pros and cons of CAMMiQ in these settings.

Response: We thank the reviewer for this suggestion. We now present the definition for F1 scores in the new **Supplementary Section 7.1**. We also newly report on the F1 scores (for species-level-all, species-level-bacteria and strain-level queries) of all tools in **Supplementary Table 5**. These are mentioned on **page 12** in the main paper: "*As can be seen in Table 3 and Supplementary Table 5, CAMMiQ achieved the best recall and F1 score (see Supplementary Section 7.1 for a definition)...*".

9. The authors should also include L2 Error in Table 3 and Table 5.

Response: We thank the reviewer for this suggestion. L2 errors have now been added to **Table 3** and **Table 5**.

10. It would be extremely helpful for interpretation of CAMMiQs superior performance via its algorithmic strategy to know where the shortest unique and doubly unique substrings are located in these genomes (including functional annotation). If these are found in mobile genetic elements vs clade-specific marker genes, etc. For well-annotated genomes in the simulated mixtures, this should be reasonably straightforward to add in.

Response: We thank the reviewer for this suggestion. We have now added two new panels to **Figure 2** (panels C and D) demonstrating the (balanced) distribution of unique and doubly-unique substrings across a number of genomes. We also discuss in detail about where such substrings are found across the genome (e.g., they are not significantly associated with genes or other functional units) in a new paragraph at the end of **page 17**: “*We further assessed whether the usage of unique and doubly-unique substrings can lead to robust genome identification and quantification performance in practice...*”.

Minor

11. Clarify: “Because any extension of a unique substring is also unique”, I think this should read “Because any extension of a shortest unique substring is also unique”. Maximal unique substrings do not exhibit this property.

Response: We have modified the definition according to the reviewer’s suggestion.

12. I strongly recommend including a comparison to MetaPalette, a k-mer painting approach to metagenomic taxonomic profiling, with a sparsity-promoting optimization procedure. Metapalette was shown to have a competitive run time compared to many of the included tools the authors included, and superior profiling performance.

Response: Unfortunately, we had difficulty in compiling MetaPalette. Even though it has a docker version, that does not allow the user to change the value of k. Nevertheless we now cite MetaPalette in the third paragraph of **page 2**: “*Unlike marker-gene based methods, k-mer based applications can use all the input reads [Reference for the MetaPalette paper].*”.

13. Ganon should be mentioned as an alternative approach that performs comparatively well to many of the tools included in this study (<https://www.ncbi.nlm.nih.gov/pmc/articles/PMC7355301/>).

Response: We now cite Ganon in the third paragraph of **page 2**: “*The large memory footprint to maintain the entire k-mer profile of each genome, for large values of k, can be reduced through hashing or subsampling the k-mers [Reference for the Ganon paper]*”.

14. Doubly unique is confusing terminology, recommend using alternative phrasing to avoid confusion.

Response: We thank the reviewer for this comment. We have considered the use of nearly-unique but that does not capture the fact that what we currently call doubly-unique substrings are those that appear exactly in two genomes. We would appreciate the reviewer's suggestions in this matter; we would consider any appropriate, alternate terminology.

15. The discussion has statements completely encapsulated in parenthesis, unclear why this is done, e.g.: (This is a simplifying assumption we employ since which genomes are represented in a query are not known in advance. In practice, the coverage for genomic sequences might be biased by GC content. We do not employ this assumption for A and A2 type queries which are more suitable for transcriptomic sequences. Our experiments on the Salmonella scRNAseq data set indeed show that these types of queries work well even though the reads are skewed by variable expression and the selection biases of single-cell technology.)

Response: We have rephrased and reformatted a variety of statements in the **Discussion** and elsewhere in the manuscript that were previously in parentheses.

References newly cited in the manuscript because of suggestions by the reviewers:

Emiola, A. & Oh, J., High throughput in situ metagenomic measurement of bacterial replication at ultra-low sequencing coverage. *Nature Communications* **9**, 4956 (2018) (reviewer 1)

Emiola, A., Zhou W. & Oh, J. Metagenomic growth rate inference of strains in situ. *Science Advances* **6**, aaz2299 (2020). (reviewer 1)

Haubold, B., Pierstorff, Möller F & Wiehe T. Genome comparison without alignment using shortest unique substrings. *BMC Bioinformatics* **6**, 123 (2006) (reviewer 2, comment 3)

Kim, D., Song, L., Breitwieser, F.P. & Salzberg, S. L. Centrifuge: rapid and sensitive classification of metagenomics sequences. *Genome Research* **26**, 172-1729 (2016) (reviewer 2, comment 2)

Lu, J., Breitwieser, F.P., Thielen, P. & Salzberg, S.L. Bracken: estimating species abundance in metagenomics data. *PeerJ* **3**, 104 (2017). (reviewer 1)

Menzel, P., Ng K. L., Krogh, A. Fast and sensitive taxonomic classification for metagenomics with Kaiju. *Nature Communications* **7**, 11257 (2016). (reviewer 2, comment 2)

Ondov, B. D., Starrett, G.J., Sappington, A., Kostic, A., Koren, S., Buck, C. B. & Phillippy, A.M. Mash screen: a high throughput sequence containment estimation for genome discovery. *Genome Biology* **20**, 232 (2019). (reviewer 1)

Philippy, A. M., Ayanbule K., Edwards, N.J. & Salzberg, S.L. Insignia: a DNA signature web server for diagnostic assay development. *Nucleic Acids Research* **37**, W229-W234 (2009) (reviewer 2, comment 2)

Piro, V. C., Dadi, T. H., Seiler, E., Reinert K. & Renard B. Y. ganon: precise metagenomics classification against large and up-to-date sets of reference sequences. *Bioinformatics* **36**(Suppl 1): i12-i20 (2020) (reviewer 2, comment 13)

REVIEWERS' COMMENTS

Reviewer #2 (Remarks to the Author):

The authors have adequately and thoroughly addressed all of my concerns, and added new experimental results that help to clarify and highlight the performance of their method.

I've been asked to comment on the revisions in response to Reviewer #1 feedback. In my view, nearly all of the original concerns of Reviewer #1 have been met. There is one outstanding major issue that needs to be addressed, however, that wasn't clearly described/delineated after revisiting the issue.

Reviewer #1 states:

> 7. With respect to performance (query time and memory usage), the maximum size data set that was included was 21.5 million reads. This is roughly the size of a MiSeq run; showing how CAMMiQ scales up to 100 million reads and beyond will help reinforce its computational efficiency/applicability to large metagenomic data sets.

The authors add the runtime info to Table 4, but leave out the memory usage. In fact, no detailed memory usage was provided in the main text as this speaks directly to the usability of the tool (it requires terabytes of RAM to run, only a limited number of researchers will be able to run it). On their GitHub page I found the following information: <https://github.com/algorithm-cancer/CAMMiQ#what-is-the-expected-computational-cost-of-cammiq>

"CAMMiQ will need to reserve up to $37 * N$ Bytes (plus the memory for maintaining the index structures) of RAM in total during its index construction step (plus certain amount of disk space)"

Given the RefSeq representative genomes DB is 150 Gbp in size (includes over 70K species), this suggests that CAMMiQ would require over 5TB of RAM to run. The authors state they use a version of RefSeq that includes 14K genomes, which should reduce the required memory down to 1.25TB or so. So in practical terms, it seems CAMMiQ will require anywhere from 1TB to over 5TB of RAM to run? This requires clarification in the discussion, including exact memory usage numbers for each step/stage of CAMMiQ to be added to the main text to help guide and orient the readers.

Specifically with respect to the following: microbial genome reference databases rapidly change over time, and will only continue their rapid growth. How does CAMMiQ's memory usage scale with respect to the size of the reference database? Are there reduced size versions of the database that would still provide superior performance that could be run in more modest resource settings? (e.g. 128GB RAM or less)

> NCBI's RefSeq database, resulting in a total of $m = 16418$ genomes.

The authors need to provide an exact version/reference to the database they used.

> Minor: Theoretical analysis leading to both L_{\min} and L_{\max} in section "Determining L_{\max} and L_{\min} " is highly controversial

In my opinion, the authors do well in their response to this comment."

Reviewers' comments:

Reviewer #2 (Remarks to the Author):

The authors have adequately and thoroughly addressed all of my concerns, and added new experimental results that help to clarify and highlight the performance of their method.

Response: We thank the reviewer for providing the useful suggestions and feedback.

I've been asked to comment on the revisions in response to Reviewer #1 feedback. In my view, nearly all of the original concerns of Reviewer #1 have been met. There is one outstanding major issue that needs to be addressed, however, that wasn't clearly described/delineated after revisiting the issue.

Reviewer #1 states:

> 7. With respect to performance (query time and memory usage), the maximum size data set that was included was 21.5 million reads. This is roughly the size of a MiSeq run; showing how CAMMiQ scales up to 100 million reads and beyond will help reinforce its computational efficiency/applicability to large metagenomic data sets.

The authors add the runtime info to Table 4, but leave out the memory usage. In fact, no detailed memory usage was provided in the main text as this speaks directly to the usability of the tool (it requires terabytes of RAM to run, only a limited number of researchers will be able to run it). On their GitHub page I found the following information:
<https://github.com/algocancer/CAMMiQ#what-is-the-expected-computational-cost-of-cammiq>

"CAMMiQ will need to reserve up to $37 * N$ Bytes (plus the memory for maintaining the index structures) of RAM in total during its index construction step (plus certain amount of disk space)"

Given the RefSeq representative genomes DB is 150 Gbp in size (includes over 70K species), this suggests that CAMMiQ would require over 5TB of RAM to run. The authors state they use a version of RefSeq that includes 14K genomes, which should reduce the required memory down to 1.25TB or so. So in practical terms, it seems CAMMiQ will require anywhere from 1TB to over 5TB of RAM to run? This requires clarification in the discussion, including exact memory usage numbers for each step/stage of CAMMiQ to be added to the main text to help guide and orient the readers.

Response: We added the memory usage in **Table 4** for all tools. We note that although the memory required by CAMMiQ index construction is relatively high as the reviewer states,

CAMMiQ supports pre built indices on commonly used databases for metagenomic studies, e.g. (the latest version of) the RefSeq bacteria, viruses and archaea database. Compared to the other tools and methods, the sizes of these pre-built indices are much smaller, due to the sparsification of unique and doubly unique substrings, allowing convenient transfer and fast downloading. We added the index size on disk in **Table 4**. We also now provide the prebuilt CAMMiQ index for all index datasets through our GitHub repo.

For memory requirements for a query, we have added the following text in the Discussion: “Another potential problem is that because CAMMiQ stores both unique and doubly unique substrings, the memory requirements would exceed those of previous software packages. We addressed this concern in some of the experiments summarized in Table 4, which show empirically that the memory requirements of CAMMiQ are comparable to those of other widely used packages and within the capabilities of currently available computers.”

Specifically with respect to the following: microbial genome reference databases rapidly change over time, and will only continue their rapid growth. How does CAMMiQ’s memory usage scale with respect to the size of the reference database? Are there reduced size versions of the database that would still provide superior performance that could be run in more modest resource settings? (e.g. 128GB RAM or less)

> NCBI's RefSeq database, resulting in a total of $m = 16418$ genomes.

The authors need to provide an exact version/reference to the database they used.

Response: We added the version of RefSeq in the **Supplementary Notes 5.1 - 5.3**, as well as in the **Data Availability** statement.

> Minor: Theoretical analysis leading to both L_{\min} and L_{\max} in section “Determining L_{\max} and L_{\min} ” is highly controversial

In my opinion, the authors do well in their response to this comment."

Response: We thank the reviewer for verifying the response. We will keep the statement in the paper.